RESEARCH COMMUNICATION

# Oxytocin promotes coordinated out-group attack during intergroup conflict in humans

**Hejing Zhang[1], Jörg Gross[2], Carsten De Dreu[2,3], Yina Ma[1]\***

[1]State Key Laboratory of Cognitive Neuroscience and Learning, International Data Group (IDG)/McGovern Institute for Brain Research, Beijing Key Laboratory of Brain Imaging and Connectomics, Beijing Normal University, Beijing, China; [2]Institute of Psychology, Leiden University, Leiden, The Netherlands; [3]Center for Research in Experimental Economics and Political Decision Making (CREED), University of Amsterdam, Amsterdam, The Netherlands

**Abstract** Intergroup conflict contributes to human discrimination and violence, but persists because individuals make costly contributions to their group's fighting capacity. Yet how group members effectively coordinate their contributions during intergroup conflict remains poorly understood. Here we examine the role of oxytocin for (the coordination of) contributions to group attack or defense in a multi-round, real-time feedback economic contest. In a double-blind placebo-controlled study with N=480 males in an Intergroup Attacker-Defender contest game, we found that oxytocin reduced contributions to attack and over time increased attacker's within-group coordination of contributions. However, rather than becoming peaceful, attackers given oxytocin better tracked their rival's historical defense and coordinated their contributions into well-timed and hence more profitable attacks. Our results reveal coordination of contributions as a critical component of successful attacks and subscribe to the possibility that oxytocin enables individuals to contribute to in-group efficiency and prosperity even when doing so implies outsiders are excluded or harmed.

**Editorial note:** This article has been through an editorial process in which the authors decide how to respond to the issues raised during peer review. The Reviewing Editor's assessment is that all the issues have been addressed (see decision letter).

DOI: https://doi.org/10.7554/eLife.40698.001

**\*For correspondence:**
yma@bnu.edu.cn

**Competing interests:** The authors declare that no competing interests exist.

## Introduction

Conflict between groups of people can profoundly change their demographic, economic and cultural outlook. Across human history, intergroup conflict functioned as a critical change agent and selection pressure that may have shaped the biological preparedness for in-group oriented cooperation and self-sacrifice on the one hand, and out-group hostility and aggression on the other hand (*Bar-Tal et al., 2007*; *Boyd and Richerson, 1982*; *Choi and Bowles, 2007*; *De Dreu et al., 2010*; *Macfarlan et al., 2014*). Out-group hostility and aggression serves to subordinate and exploit rivaling out-groups (henceforth out-group attack), and/or to defend the in-group against such hostility from neighboring groups (henceforth in-group defense) (*De Dreu et al., 2016a*; *Halevy et al., 2008*; *Rusch, 2014a*; *Rusch, 2014b*; *Lopez, 2017*; *Wrangham, 2018*). Recent work has showed that individuals are more strongly motivated to contribute to defend one's own group than to attack out-groups and, importantly, that out-group attacks frequently fail because individual contributions to out-group attacks are poorly coordinated (*De Dreu et al., 2016a*). Indeed, attacking rivals who are expected to have strong defenses more likely results in failure and waste, than attacking groups

**eLife digest** Conflict between groups is a recurring theme in human history. We tend to form social bonds with others who share the same characteristics as ourselves, whether that is nationality, ethnicity, or supporting the same football team. Individuals that belong to the same group as us comprise our 'in-group'. All other individuals make up our 'out-groups'. Competition and conflict with out-groups – from benign sporting rivalry to warfare – has a key role in shaping human cultures and societies.

Such conflict often requires individuals to act in ways that harm their own self-interests. It also requires them to coordinate their actions with other members of their in-group. How does our biology drive this behavior? When small groups prepare for conflict with other groups, they often perform social bonding routines and rituals. These trigger the brain to release a hormone called oxytocin into the bloodstream. Known as the 'love hormone', oxytocin helps promote pair bonding as well as social bonding with in-group members. Studies in both humans and monkeys show that boosting oxytocin levels artificially via a nasal spray makes individuals more trusting and cooperative.

But Zhang et al. now show that the 'love hormone' also helps individuals launch more coordinated 'attacks' on out-groups. In a study involving a multi-round economic contest game between groups of 'attackers' and 'defenders', oxytocin did not make attackers less aggressive. Instead it enabled them to better coordinate their attacks. Each contest game involved three attackers individually contributing money to a group pool to outbid the other group and win more money, and three defenders making similar contributions to their own group pool to defend it against the rivals' attacks and protect themselves from losing all their money. Attackers who used an oxytocin nasal spray were better at tracking their rivals' defensive strategies than attackers whose nasal spray contained a placebo. Under the influence of oxytocin, the attackers timed their strikes to occur when their rivals were vulnerable. Over time, the oxytocin users became better at coordinating their behavior with other members of their in-group. This resulted in more earnings.

Success – and even survival – in intergroup conflicts depends on how willing individuals are to make contributions that incur a personal cost. They also depend on how well individuals coordinate their contributions. Social strategies, such as leading by example, and neurobiological mechanisms such as oxytocin can both help achieve the coordination needed to exploit out-group rivals.
DOI: https://doi.org/10.7554/eLife.40698.002

expected to be weak and defenseless (*De Dreu et al., 2016b*; *Grossman and Kim, 2002*; *Goeree et al., 2003*). Thus, to succeed in intergroup competition and conflict, group members not only need to contribute to their group's competitive strength, but they also need to coordinate within their group the intensity and timing of attacks.

How group members coordinate behavior into effective joint actions during intergroup competition and conflict remains poorly understood (*De Dreu et al., 2016a*) and we lack insight into the underlying neurobiological mechanism. Here, we target the neuro-hormone oxytocin as a neurobiological mechanism underlying within-group coordination of out-group attack in dynamic intergroup contests. Studies in ethnography and anthropology have shown that small groups preparing for intergroup conflict build cohesion and commitment through social bonding routines and rituals. Groups selectively invite friends to join a raid (*Glowacki et al., 2016*), bond like family (*Macfarlan et al., 2014*), and engage in cultural rituals that simulate self-sacrifice, cooperation and coordination (*Xygalatas et al., 2013*). Neuroendocrine studies (*Burkett et al., 2016*; *Carter, 2014*; *Ma et al., 2016b*; *Samuni et al., 2017*) have linked these and related practices to the release of oxytocin, a nine-amino acid neuropeptide. In turn, intranasal administration of oxytocin has been shown to modulate neural responses in brain regions involved in threat detection and reward processing (*Ma et al., 2016b*; *Paloyelis et al., 2016*; *Wang et al., 2017*; *Liu et al., 2019*). Moreover, in both non-human and human primates, elevated levels of oxytocin via intranasal administration have been linked to a range of cognitive and behavioral effects including within-group conformity, trust, affiliation, and cooperation (*Arueti et al., 2013*; *Aydogan et al., 2017*; *De Dreu et al., 2010*;

*Madden and Clutton-Brock, 2011*; *Rilling et al., 2012*; *Samuni et al., 2017*; *Stallen et al., 2012*; *Yan et al., 2018*).

Whereas these work together point to oxytocin as a possible neurobiological mechanism underlying group coordination, three issues remain unclear. First, we lack empirical evidence for the possibility that oxytocin promotes group-level coordination of collective action in general, and during intergroup competition and conflict in particular. Second, we poorly understand how group members coordinate collective action during intergroup competition and conflict, and whether oxytocin can directly influence coordination and/or the strategy group members use to coordinate. Third, we do not know whether oxytocin differentially modulates tacit coordination within groups when collective contributions are focused on attacking one's rival, versus defending the in-group against possible attacks by one's rival.

To address these issues, we examined the role of oxytocin in 80 interactive, multi-round contests between three-person attacker and three-person defender groups. Group members on each side made individual contributions to their group pool (aimed at attacking the other side, or at defending against such possible attacks). We provided individuals with real-time feedback on group investments and success after each contest round, which allowed group members to learn and adapt to their rival's past investments, and use the history of play as a focal point to coordinate their future attacks and defenses. Indeed, when lacking explicit coordination mechanisms such as a leader or decision-making protocols (*De Dreu et al., 2016a*; *Gavrilets and Fortunato, 2014*; *Hermalin, 1998*), groups use social norms and focal points to tacitly coordinate collective action (*Halevy and Chou, 2014*; *Schelling, 1960*). In multi-round intergroup contests, the rival's history of play can serve as a focal point for groups to coordinate their contributions to attack and/or defend (*De Dreu et al., 2016b*). Thus, when attacking out-groups, group members may coordinate their contributions on their rival's historical level of defense, and attack when historical defense is low and the target appears vulnerable; and not attack when historical defense is high and the target appears strong and difficult to beat.

We expected effects of oxytocin to be stronger during out-group attack than during in-group defense. In-group defense is first and foremost an adaptation to the attackers' (past) aggression and is typically well-coordinated—individuals fight towards the same goal of self-preservation and group survival. Although there is some evidence that oxytocin may increase protective aggression (for a review see *De Dreu and Kret, 2016*), administering oxytocin may contribute little to the relatively high baseline levels of contribution and coordination during in-group defense. In contrast, out-group attacks are typically less strong, more variable, and less well-coordinated (*De Dreu et al., 2016a*), leaving more room for oxytocin administration to influence behavior. We anticipated two possibilities for the effect of oxytocin. On the one hand, oxytocin has been linked to prosociality (*Madden and Clutton-Brock, 2011*; *Rilling et al., 2012*; *Liu et al., 2019*), empathy (*Hurlemann et al., 2010*; *Bartz et al., 2010*) and consolation (*Burkett et al., 2016*). This line of research would lead us to anticipate that oxytocin may enable individuals in attacker groups to coordinate on a peaceful no-attack strategy that is independent of the rivals' history of defense. On the other hand, oxytocin has been shown to enhance in-group conformity (*Aydogan et al., 2017*; *Stallen et al., 2012*; *De Dreu and Kret, 2016*), increase behavioral coordination and neural synchronization within pairs of individuals performing a joint task (*Arueti et al., 2013*; *Mu et al., 2016*), enhance facial mimicry, motor imitation, and neural responses linked to action-intention mirroring (*Korb et al., 2016*; *De Coster et al., 2014*; *Kret and De Dreu, 2017*; *Levy et al., 2016*), and improve memory and learning from feedback (*Striepens et al., 2012*; *Guastella et al., 2008*; *Ma et al., 2016a*). These functionalities alone and in combination may facilitate coordination on social norms and shared focal points that emerge during group interaction, suggesting that oxytocin may facilitate the use of rival's history, enable individuals in attacker groups to coordinate better on the level and timing of their attacks, and to efficiently appropriate resources from their out-group. Evidence for this possibility would be consistent with earlier findings that oxytocin shifts the focus from self-interest to in-group interests (*De Dreu and Kret, 2016*), limits trust and cooperation to in-group members (*De Dreu et al., 2010*; *Ma et al., 2015*; *Ten Velden et al., 2017*), and promotes aggression toward threatening rivals (*Madden and Clutton-Brock, 2011*; *Samuni et al., 2017*; *Striepens et al., 2012*; *Ne'eman et al., 2016*).

# Results

We examined these possibilities using a dynamic, fully incentivized Intergroup Attacker-Defender Contest game (*De Dreu et al., 2016a*). The IADC is an all-pay contest (*Abbink et al., 2010*; *Dechenaux et al., 2015*) involving six individuals randomly assigned to a three-person attacker and a three-person defender group. In the IADC game, 3 attackers made individual contributions to the group pool to subordinate the other group and increase gains through victory (but always keep the remaining resources), and 3 defenders contributed to their group pool to defend rival's attack and protect against loss and defeat (defenders 'survive' from left with 0 only if defense succeeds). We created 80 IADC sessions (with *N* = 480 healthy males); in 40 (40) sessions, participants received intranasal oxytocin (matching placebo) (*Figure 1*). Each IADC session involved two blocks of 15 investment rounds with real-time feedback in between rounds. For each investment round (*Table 1*), each individual received an initial endowment of 20 MUs (Monetary Units). Each individual decided the amount ($I_i$, $0 \leq I_i \leq 20$) to the group's pool $G$ ($0 \leq G \leq 60$, $G_{Attacker} = I_{Attacker-1} + I_{Attacker-2} + I_{Attacker-3}$, $G_{Defender} = I_{Defender-1} + I_{Defender-2} + I_{Defender-3}$). Contributions were wasted, but when $G_{Attacker} \leq G_{Defender}$, attackers failed and defenders survived and all six individuals kept their remaining endowment (leftovers, $20 - I_i$). However, when $G_{Attacker} > G_{Defender}$, defenders failed and left with 0. The attacker group won and took away defender group's remaining MU (spoils from winning, $60 - G_{Defender}$), which were divided equally among three attacker group members (each attacker member received: $(60 - G_{Defender})/3$) and added to their remaining endowments ($20 - I_{Attacker-i}$). Thus, contributions in the attacker group ($G_{Attacker}$) and in the defender group ($G_{Defender}$) reflect the contribution level to out-group attack and to in-group defense, respectively. In one 15-round block, individuals made their decisions individually and simultaneously without communication (i.e. Simultaneous decision-making block). In the other 15-round block (order counterbalanced), individuals within groups made their decisions in sequence: one randomly drawn member made his decision first, followed by the second randomly selected member who was informed about the first member's contribution, and then the third and final member made his decision (i.e. Sequential decision-making block, *De Dreu et al., 2016a*; *Gavrilets and Fortunato, 2014*; *Figure 2*).

The sequential decision-making protocol acts as a 'coordination device' that facilitates behavioral coordination within groups (*De Dreu et al., 2016a*; *Gavrilets and Fortunato, 2014*; *Hermalin, 1998*). This treatment thus provides a benchmark to compare groups in the simultaneous decision-protocol who lack an explicit coordination mechanism and have to find other means—such as a shared social norm or the rival's past defense — to coordinate individual contributions into effective joint action (*De Dreu et al., 2016a*). We operationalized within-group coordination as behavioral alignment of contribution to the group, indicated by the variance in contributions to group fighting,

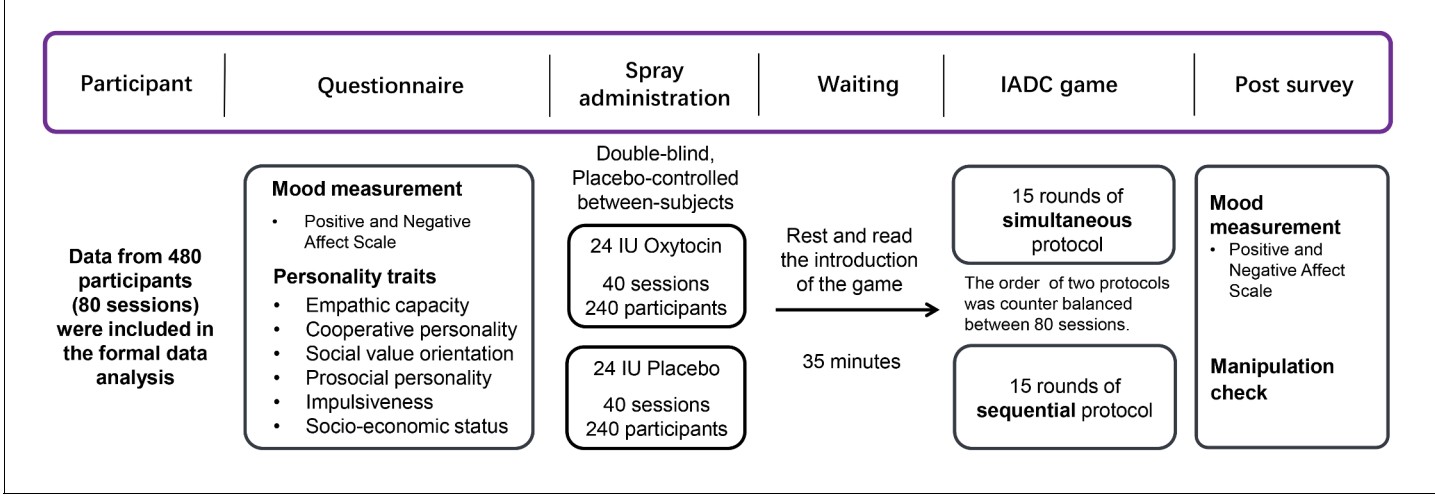

**Figure 1.** General experimental procedure.
DOI: https://doi.org/10.7554/eLife.40698.003

**Table 1.** Payoff matrix of one-round IADC.

| Role | Participant | Initial endowment (MU) | Individual contribution ($I_i$) | Group pool (G) | Payment — Attackers lose $G_{Attacker} \leq G_{Defender}$ — Leftover | Payment — Attackers win $G_{Attacker} > G_{Defender}$ — Leftover | Payment — Attackers win $G_{Attacker} > G_{Defender}$ — Spoil |
|---|---|---|---|---|---|---|---|
| Attack | Attacker-1 | 20 | $I_{Attacker-1}$ | $G_{Attacker}$ | $20 - I_{Attacker-1}$ | $20 - I_{Attacker-1}$ | $(60 - G_{Defender})/3$ |
| | Attacker-2 | 20 | $I_{Attacker-2}$ | | $20 - I_{Attacker-2}$ | $20 - I_{Attacker-2}$ | $(60 - G_{Defender})/3$ |
| | Attacker-3 | 20 | $I_{Attacker-3}$ | | $20 - I_{Attacker-3}$ | $20 - I_{Attacker-3}$ | $(60 - G_{Defender})/3$ |
| Defend | Defender-1 | 20 | $I_{Defender-1}$ | $G_{Defender}$ | $20 - I_{Defender-1}$ | 0 | 0 |
| | Defender-2 | 20 | $I_{Defender-2}$ | | $20 - I_{Defender-2}$ | 0 | 0 |
| | Defender-3 | 20 | $I_{Defender-3}$ | | $20 - I_{Defender-3}$ | 0 | 0 |

Table note: For each round, each individual received an initial endowment of 20 MUs (Monetary Units).

Each individual decided the amount ($I_i$, $0 \leq I_i \leq 20$) to the group's pool G ($0 \leq G \leq 60$, $G_{Attacker} = I_{Attacker-1} + I_{Attacker-2} + I_{Attacker-3}$, $G_{Defender} = I_{Defender-1} + I_{Defender-2} + I_{Defender-3}$). When $G_{Attacker} \leq G_{Defender}$, attackers failed and defenders survived and all six individuals kept their remaining endowment (leftovers, $20 - I_i$). When $G_{Attacker} > G_{Defender}$, defenders failed and left with 0. The attackers won and took away defenders' remaining MU (spoils from winning, $60 - G_{Defender}$), which were divided equally among attacker group members (each attacker: $(60 - G_{Defender})/3$) and added to their remaining endowments ($20 - I_{Attacker-i}$).

DOI: https://doi.org/10.7554/eLife.40698.005

with lower group-level variance reflecting better coordination. We expected that effects of oxytocin on group coordination would emerge especially under simultaneous rather than sequential decision-making.

Before examining within-group coordination, we examined treatment effects on contributions to group pool. Individual contributions were averaged across the three members within each three-person group, and submitted to a 2 (Treatment: Oxytocin vs. Placebo) × 2 (Role: Attack vs. Defense) × 2 (Procedure: Simultaneous vs. Sequential) × 15 (Rounds) mixed-model Analysis of Variance (ANOVA) with Treatment between-sessions. Individuals contributed more under sequential than simultaneous decision-making ($M \pm SE = 6.89 \pm 0.20$ vs. $6.47 \pm 0.24$; $F(1, 78) = 5.109$, $p = 0.027$, $\eta^2 = 0.061$), and somewhat less when given oxytocin rather than placebo ($M \pm SE = 6.28 \pm 0.27$ vs. $7.08 \pm 0.30$; $F(1, 78) = 3.719$, $p = 0.057$, $\eta^2 = 0.046$; marginal significance). *Figure 3A* showed higher contributions to in-group defense than to out-group attack, especially in earlier rounds (Role, $F(1, 78) = 287.903$, $p < 0.001$; $\eta^2 = 0.787$; Role × Round, $F(14, 65) = 4.529$, $p < 0.001$, $\eta^2 = 0.494$).

Treatment effects also emerged when we examined the number of non-contributors. There were more non-contributors in attacker compared to defender groups ($M \pm SE = 20.23 \pm 0.90$ vs. $4.25 \pm 0.44$; $F(1, 78) = 408.489$, $p < 0.001$; $\eta^2 = 0.840$), and more non-contributors in groups given oxytocin than placebo ($M \pm SE = 13.46 \pm 0.90$ vs. $11.01 \pm 0.75$; $F(1,78) = 4.345$, $p = 0.040$; $\eta^2 = 0.053$). Crucially, oxytocin increased the number of non-contributors in attacker groups but not in defender groups (Role × Treatment, $F(1, 78) = 5.043$, $p = 0.028$, $\eta^2 = 0.061$, *Figure 3B*). This Role × Treatment effect is especially true when decisions were made simultaneously ($F(1, 78) = 5.712$, $p = 0.019$, $\eta^2 = 0.068$) but less so when decisions were made sequentially ($F(1, 78) = 2.143$, $p = 0.147$; $\eta^2 = 0.027$). We then examined participants' decision time when deciding to (not) contribute, we showed that individuals in attacker groups made their decisions to not contribute faster than to contribute ($F(1, 78) = 137.679$, $p < 0.001$, $\eta^2 = 0.641$). Oxytocin increased the speed with which individuals in attacker groups decided to not contribute (Treatment × Contribute: $F(1, 77) = 4.857$, $p = 0.031$; $\eta^2 = 0.059$, *Figure 4*).

Next, we analyzed group-level coordination operationalized as the within-group variance in contributions, with lower variance indicating stronger coordination (*De Dreu et al., 2016a*). First, there was a Procedure × Role interaction ($F(1, 78) = 101.978$, $p < 0.001$, $\eta^2 = 0.567$) showing that sequential decision-protocol facilitated coordination for attack ($F(1, 78) = 77.852$, $p < 0.001$, $\eta^2 = 0.500$) and, unexpectedly, reduced coordination for defense ($F(1, 78) = 39.268$, $p < 0.001$, $\eta^2 = 0.335$). Importantly, as predicted, oxytocin facilitated group-level coordination (i.e. reduced within-group variance in contributions) of out-group attack but not of in-group defense, especially in earlier

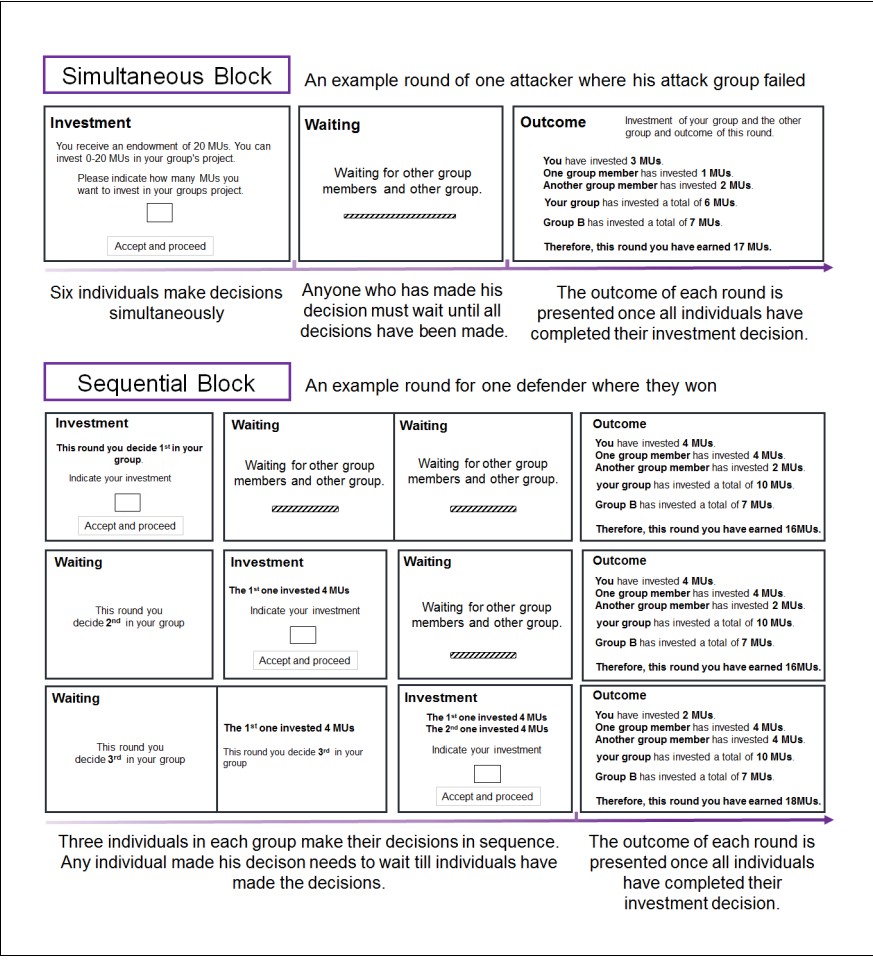

**Figure 2.** Illustration of one round of the IADC game in the simultaneous and sequential decision-making blocks, respectively.
DOI: https://doi.org/10.7554/eLife.40698.004

rounds (Role × Treatment × Round, $F(14, 1092) = 1.753$, p = 0.041, $\eta^2 = 0.022$; *Figure 5*A; *Figure 5—figure supplement 1* for similar results of another index of coordination).

To examine what strategy attackers given oxytocin coordinated on, we first examined the contest outcomes. As noted, oxytocin may enable groups to coordinate on a peaceful 'no-attack strategy', in which case we should find (1) lower victories in attacker groups given oxytocin rather than placebo, (2) the oxytocin effect on the number of non-contributing attackers should not differ when attacks succeed or fail, and (3) no effect of oxytocin on tracking the rival's defense history. This was not the case. First, the number of victories was similar in attacker groups given oxytocin (M ± SE: 24.8 ± 1.8%) and placebo (M ± SE: 26.6 ± 1.6%), ($F(1, 78)=0.572$, p=0.452; $\eta^2 = 0.007$). Thus, rather than making groups coordinate on a peaceful no-attack strategy, oxytocin may enable groups to coordinate on attacking at the right moment with the right force. Indeed, analyses of the spoils and leftovers showed that groups given oxytocin rather than placebo had higher leftovers when attacks failed ($t(78) = 2.609$, p=0.011, Cohen's d = 0.581, 95% *Confidence Interval* (CI), 0.232 to 1.76, *Figure 5B*) and somewhat higher spoils when attacks were successful ($t(74) = 1.819$, p=0.073, Cohen's d = 0.419, 95% CI, −0.086 to 1.888, marginal significance, *Figure 5C*). To illustrate the increased efficiency of attack under oxytocin, attackers' contribution and payment under the four conditions (i.e. Simultaneous/Sequential x Oxytocin/Placebo) were plotted in *Figure 5D* using a bootstrapping technique (*Davison and Hinkley, 1977*). Specifically, a bootstrapped dataset with sample size $N = 40$ was resampled with replacement separately for each of the four conditions. The mean contribution and payment of the bootstrapped sample was then calculated and saved as a

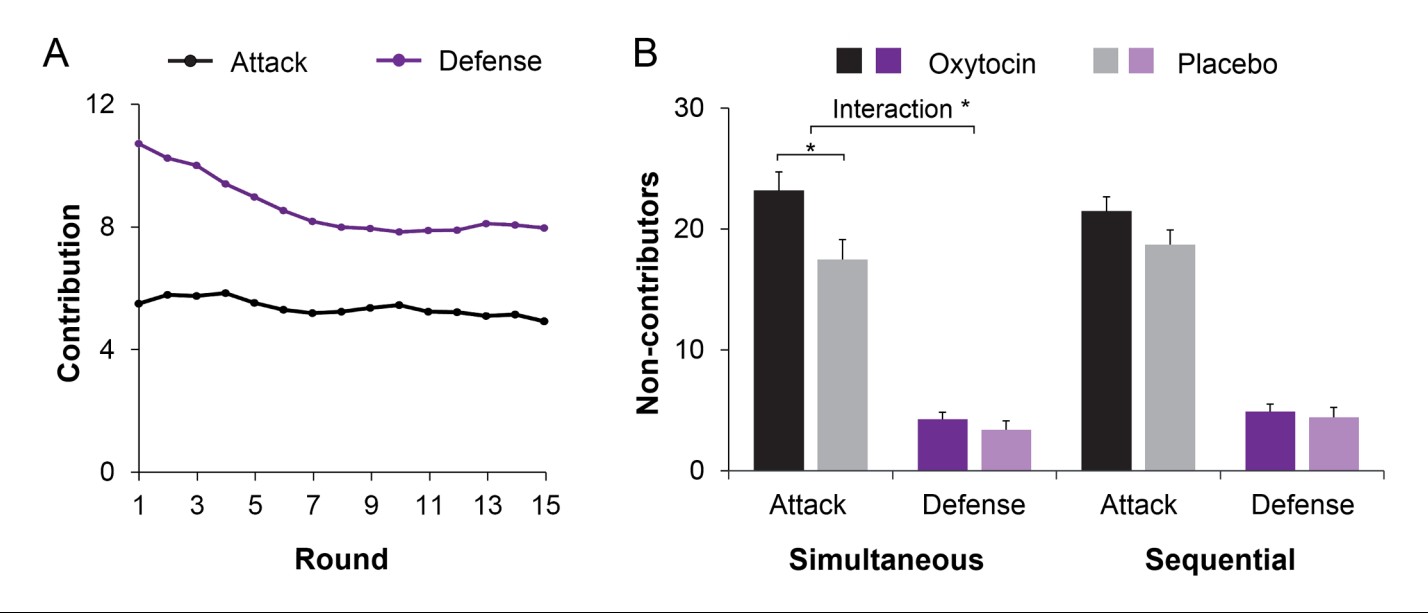

**Figure 3.** Oxytocin modulates contributions to group fighting. (**A**) Attackers contribute less than defenders, especially in early rounds (range 0–20). Curves were smoothed with a moving average window of three investment rounds. (**B**) Giving individuals oxytocin rather than placebo increases the number of non-contributing members in attacker groups especially under simultaneous decision-making (with 0–3 members per round across 15 rounds; range 0–45; displayed M ± 1 SE). Connectors indicate significant difference, with *p<0.05.

DOI: https://doi.org/10.7554/eLife.40698.006

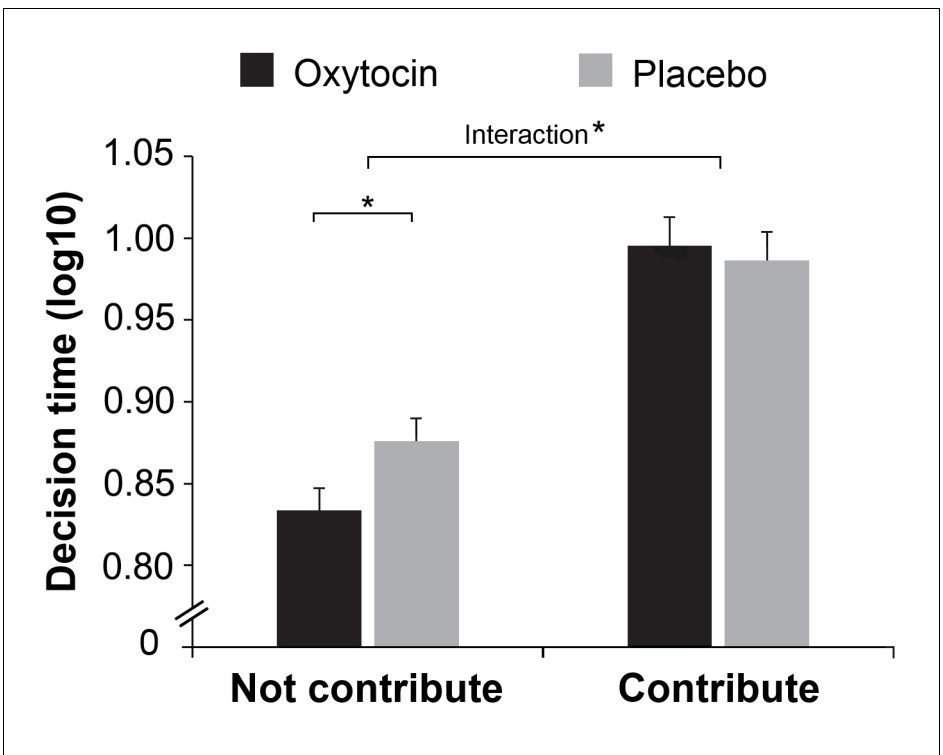

**Figure 4.** Response time for decisions to (not) contribute. Oxytocin increased the speed with which attackers made their decisions to not contribute. Connectors indicate significant difference, with *p<0.05.

DOI: https://doi.org/10.7554/eLife.40698.007

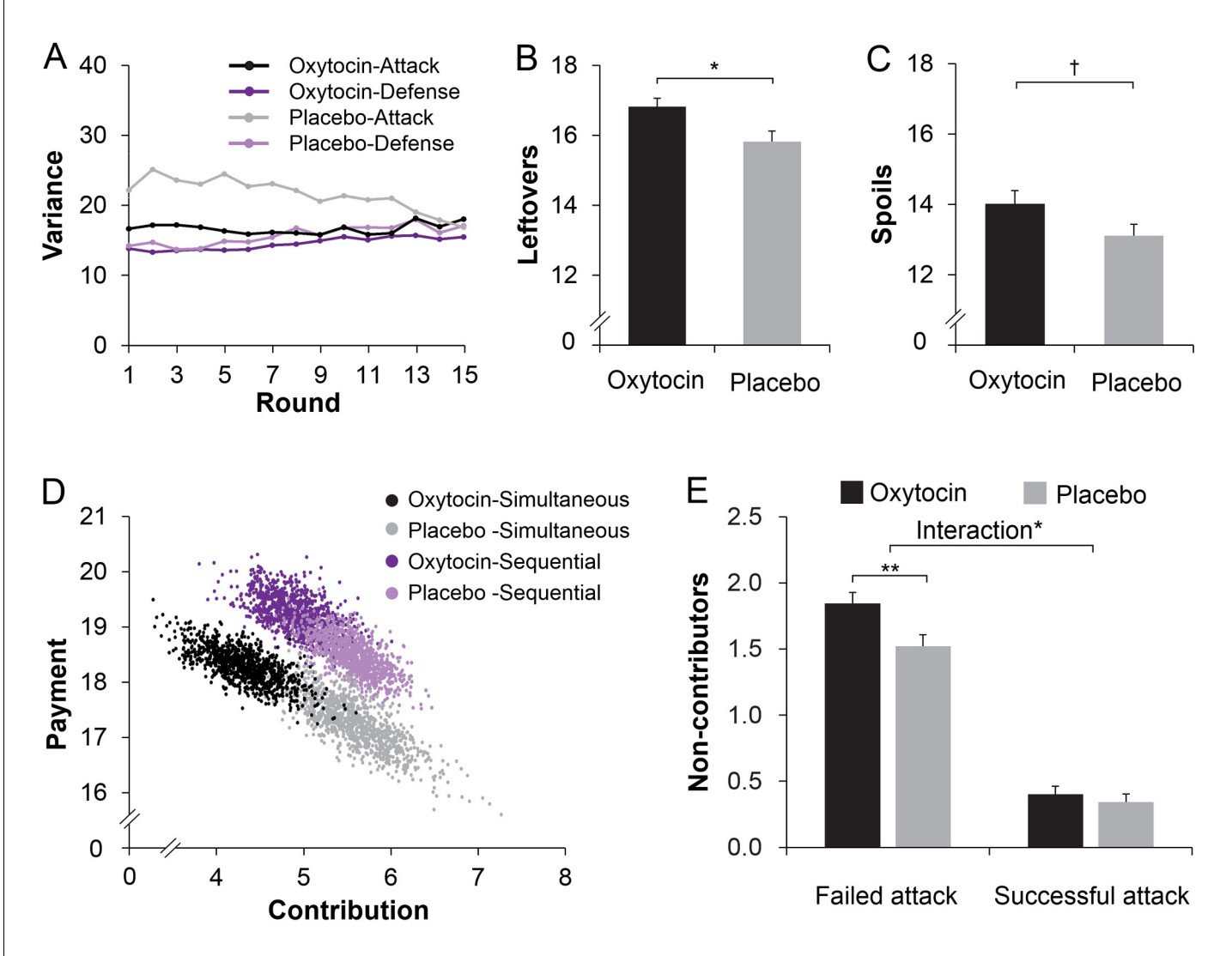

**Figure 5.** Oxytocin modulates within-group coordination. (A) Giving individuals oxytocin rather than placebo enables better coordination (lower within-group variance) in attacker groups, especially in early rounds. Curves were smoothed with a moving average window of three investment rounds. (B/C) Giving attackers oxytocin rather than placebo increases their leftovers when not winning the conflict (B) and spoils from winning conflicts (C) (N = 76 because four attacker groups never won). (D) Bootstrapping illustration of the oxytocin shifts on the contribution and payment. Bivariate distributions of 1000 bootstrapped sample means for each condition (Treatment x Procedure) plotted against the contribution and payment. (E) Oxytocin increased non-contributing attackers only in failed attacks but not in successful attacks. Connectors indicate significant difference, with † p < 0.10; *p < 0.05; **p < 0.01.

DOI: https://doi.org/10.7554/eLife.40698.008

The following figure supplement is available for figure 5:

**Figure supplement 1.** Oxytocin increases attacker group's within-group coordination especially in the simultaneous decision-making block. Connectors indicate significant difference, with *p<0.05, *** p< 0.001.
DOI: https://doi.org/10.7554/eLife.40698.009

new data point. For each condition, this procedure was repeated for 1000 times to represent the population information. As illustrated in *Figure 5D*, oxytocin decreased contributions to attack (a shift toward less contribution) but increased payment (a shift toward more payment). The distribution of attacker groups under oxytocin and placebo in a space defined by the two vectors (contribution and payment) indicates a clear separation of oxytocin and placebo treatments (especially when decisions were made simultaneously, that is black dots vs. grey dots).

Second, we conducted an ANOVA on the average number of non-contributing attackers, with Treatment (Oxytocin vs. Placebo) as a between-subjects factor, Procedure (Simultaneous vs. Sequential) and Success (Success vs. Failure) as within-subjects factors. There were more non-contributing attackers when attack failed than succeeded (Failure vs. Success: M ± SE = 1.68 ± 0.60 vs. 0.37 ± 0.05, F (1, 74) = 636.941, p < 0.001; $\eta^2$ = 0.891). Interestingly, we found a significant Treatment ×2 Success interaction on the number of non-contributing attackers (F (1, 74) = 6.345, p = 0.014; $\eta^2$ = 0.075, *Figure 5E*): Oxytocin increased the number of non-contributing attackers only in failed rounds (Oxytocin vs. Placebo: M ± SE = 1.84 ± 0.09 vs. 1.52 ± 0.09; F (1, 74) = 7.036; p = 0.010; $\eta^2$ = 0.083) but not in successful attacks (Oxytocin vs. Placebo: M ± SE = 0.34 ± 0.06 vs. 0.40 ± 0.06; F (1, 74) = 0.448; p = 0.505; $\eta^2$ = 0.006).

Third, we examined how attacker groups collectively identified when to attack by creating a past defense parameter α (average defender group's investment in the last two rounds, that is ($D_{j-1}$+$D_{j-2}$)/2 on round j) and regressed attacker group's investments onto α (attack increased when defender groups were vulnerable rather than strong, as indicated by α approaching -1). It showed that attacker groups given oxytocin tracked their rival's past defense and attacked especially when defenders appeared more rather than less vulnerable (i.e. attack regressed *negatively* on rival's historical defense). Specifically, when decisions were made simultaneously, attack regressed more strongly on α when groups received oxytocin (M ± SE: -0.30 ± 0.05) rather than placebo (M ± SE: -0.042 ± 0.098; t(78) = -2.334; p = 0.022, Cohen's d = -0.522, 95% CI, -0.482 to -0.038), and under oxytocin but not placebo, the regression on α was also stronger in simultaneous rather than sequential decision-making (Treatment × Procedure interaction, *F*(1, 78) = 8.312, *p* = 0.005, $\eta^2$ = 0.097, *Figure 6A*).

Combined, results suggest that groups given oxytocin created more spoils from winning and had higher leftovers when attacks failed because they better coordinated attack at the right time and with the proper force. Indeed, when decisions were made simultaneously, the more strongly attacker groups relied on tracking parameter α, the lower their within-group variance when contributing (*r* = 0.281, p=0.012, *Figure 6B*), and the lower within-group variance when contributing, the higher the attacker's spoils when winning the conflict (*r* = −0.328, p=0.004, *Figure 6C*). Indirect mediation analyses confirmed that the attacker's higher spoils under oxytocin was mediated by (i) increased tracking of defender group's past investments and (ii) concomitant increased within-group coordination (indirect effect = 0.147, SE = 0.103; 95% CI, 0.019 to 0.511, *Figure 6D*).

## Discussion

To be victorious in intergroup conflict, group members not only need to contribute to their group's fighting capacity. They also need to coordinate collective action so that they attack when their rival is expected to be weak, and avoid wasting resources on attacking tough defenders. Here we found, using a dynamic intergroup contest between attackers and defenders, that those groups who tracked their defender's history of play and coordinated their attacks of weak rather than strong defenders wasted less resources on failed attacks and enjoyed greater spoils when winning. In addition, we uncovered that oxytocin serves as a neurobiological mechanism underlying such well-timed and coordinated attacks. Specifically, we found that oxytocin enabled individuals within attacker groups to converge their individual contributions on each other more, to collectively refrain from attacking apparently strong defenders, and effectively attacking weak defenders. These findings emerged when groups lack an explicit coordination device; providing attacker groups with a sequential decision-making protocol as an explicit coordination device substituted oxytocin-induced tacit coordination.

Our finding that oxytocin enables within-group coordination of contributions to out-group attacks resonates with two heretofore disconnected sets of findings—that small groups of warriors engage in social bonding and cultural rites (*Glowacki et al., 2016*; *Macfarlan et al., 2014*; *Schelling, 1960*; *Wilson and Wilson, 2007*; *Xygalatas et al., 2013*), and that social bonding and synchronized action can trigger the release of oxytocin (*Burkett et al., 2016*; *Carter, 2014*; *Samuni et al., 2017*). Combined with the current results, it suggests that oxytocin might be a potential neurobiological mechanism through which social bonding and performing coordinated rituals can help groups to better coordinate their attacks. Our findings furthermore suggest that such oxytocin-mediated within-group coordination can be substituted by institutional arrangements, such as a sequential decision-making

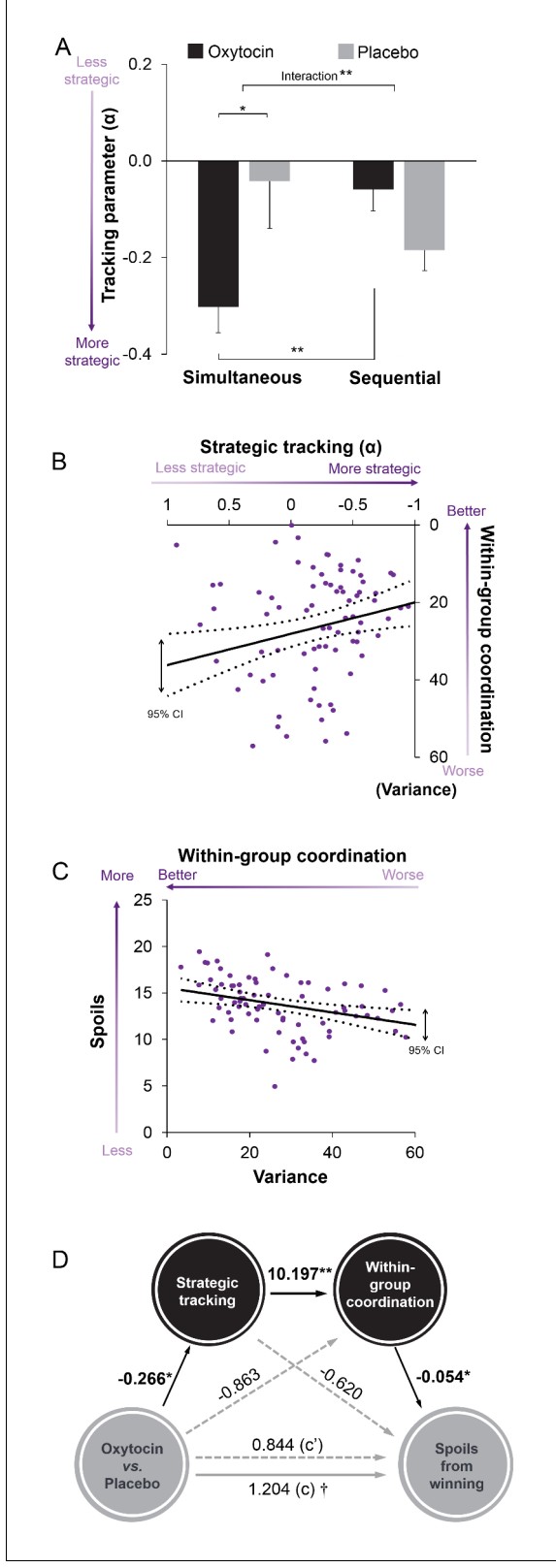

**Figure 6.** Oxytocin enables a track-and-attack strategy (strength of attack increases when defender groups are vulnerable rather than strong, as indicated by $\alpha \to -1$). (**A**) When attacker groups are given oxytocin investments regress negatively on $\alpha$ (the rival's historical investments to defense), especially during simultaneous than sequential decision-making. (**B**) Stronger negative regression of attack on rival's defense history ($\alpha \to -1$) among

*Figure 6 continued on next page*

*Figure 6 continued*

attacker groups associates with better coordination (i.e. lower within-group variance). (C) Better coordination (i.e. lower within-group variance) associates with higher spoils when winning the conflict. (D) Oxytocin's effect on spoils from successful attacks is mediated by treatment effects on tracking $\alpha$ (more strategic when $\alpha \rightarrow -1$) and better within-group coordination. † $p < 0.10$; * $p < 0.05$; ** $p < 0.010$).

DOI: https://doi.org/10.7554/eLife.40698.010

The following figure supplement is available for figure 6:

**Figure supplement 1.** Oxytocin influences payment through its effects on strategic tracking and better within-group coordination.

DOI: https://doi.org/10.7554/eLife.40698.011

protocol. We speculate that related institutional arrangements, like appointing a leader or having open communication channels can similarly obviate the need for bonding rituals and/or oxytocin to make out-group attack well-coordinated and successful.

Neither oxytocin nor the sequential decision-protocol contributed to within-group coordination in defender groups. One possibility is that in-group defense is tacitly well-coordinated because of the stronger alignment of individual interests within defender groups. After all, when defender groups survive an enemy attack, those who did not contribute to in-group defense come out relatively wealthy. But when in-group defense fails, all group members lose regardless of whether they contributed or not. As a result of this stronger common fate in defender groups, individuals contribute spontaneously and tacitly coordinate well on avoiding collective defeat. Exogenous enhancers, whether at the neurobiological or institutional level, appear not necessary.

Previous studies provided evidence for the role of oxytocin in intergroup interaction: It can increase the positive view and benign approach of the in-group and has been linked to subtle forms of intergroup discrimination (*De Dreu et al., 2010*; *De Dreu and Kret, 2016*; *Ma et al., 2015*; *Stallen et al., 2012*; *Ten Velden et al., 2017*). Without exception, this early evidence was limited to individual-level decision-making, and did not clearly distinguish the distinct motives for contributing to in-group efficiency and out-group hostility. It remained unknown whether and how oxytocin differentially affects attack and defense behavior during intergroup conflict. Using a dynamic attack-defense contest we revealed selective effects of oxytocin on group attacking. As such, the present study provides a new perspective on oxytocin in intergroup conflict by highlighting its functionality for strategic attack (rather than defense). In doing so, we also obtained first-time evidence that in-group coordination for collective action can be tracked to evolutionary preserved neurobiological factors.

Our study has two potential limitations. First, it involved only Chinese males as study participants. Whereas we cannot exclude specific cultural effects, findings for the comparison between out-group attack and in-group defense and between sequential and simultaneous decision-making protocols resonate with findings obtained in Western culture, with both male and female participants (*De Dreu et al., 2016a*). This generates confidence in the generality of the behavioral effects for attack/defense and decision protocols. Furthermore, recent work on the role of oxytocin in in-group cooperation suggests similar effects for both male and female participants (*Ten Velden et al., 2017*). Thus, and given the absence of strong counter-evidence, we cautiously conclude that current findings may generalize across cultural contexts and apply to male as well as female participants. Second, it is unclear how exogenous administration of oxytocin operates at the neurophysiological level. Whereas some have questioned whether intranasal administration of oxytocin can have a direct impact on brain and behavior (*Leng and Ludwig, 2016*), recent studies in rodents (*Neumann et al., 2013*; *Tanaka et al., 2018*), rhesus macaques (*Lee et al., 2018*; *Modi et al., 2014*) and in humans (*Paloyelis et al., 2016*) collectively suggest that intranasal oxytocin elevates brain-level presence of oxytocin, and impacts behavioral decisions through neural networks involved in threat detection and reward processing (*Carter, 2014*; *Hurlemann et al., 2010*; *Ma et al., 2016b*; *Rilling et al., 2012*; *Stallen et al., 2012*; *Wang et al., 2017*; *Liu et al., 2019*). In addition, there is some evidence that higher levels of oxytocin in saliva, blood, or urine relate to in-group affiliation and cooperation (*Madden and Clutton-Brock, 2011*; *Samuni et al., 2017*) and intergroup discrimination (*Levy et al., 2016*). Nevertheless, new research is needed to uncover the neurophysiological pathways through

which intranasal oxytocin impact human cognition and behavior, and how findings on intranasal oxytocin relate to endogenous oxytocin measured from saliva, urine, or blood samples.

Intergroup competition and conflict shape the economic and cultural outlook of groups and societies, and interferes with individual life-trajectories. Success and survival in times of intergroup competition and conflict depend on the extent to which individuals make personally costly contributions and, as shown here, on their ability to coordinate their contributions when attacking out-groups, or defending against threatening out-groups. Both institutional arrangements, such as leading-by-example, and neurobiological mechanisms, such as oxytocin, facilitate behavioral coordination and the effective exploitation of out-group rivals. Indeed, as shown here, providing group members with oxytocin or an explicit coordination device enables them to waste less on failed attacks, and to earn more from their victories.

## Materials and methods

### Participants

We recruited 486 healthy males, mostly science and engineering students, as paid volunteers. One IADC session ($N$ = 6) was dropped from data analysis because of technical failure. Data from 480 participants (80 IADC sessions, age 18–29 years; $M \pm SE$ = 20.28±0.09 years) were included in the final data analysis. The data analysis on the current study was conducted on a 6-person-group level, thus we conducted sample size estimation by G*Power to determine the number of groups sufficient to detect a reliable effect. Based on an estimated average small-to-medium effect size of oxytocin effect on social behaviors (d = 0.28, *Walum et al., 2016*), 80 6-person groups were needed to detect a significant effect ($\alpha$ = 0.05, $\beta$ = 0.85, ANOVA: repeated measures, within-between interaction, G-Power, *Faul et al., 2009*). All participants were healthy and had normal or corrected-to-normal vision and no history of neurological or psychiatric disorders. Those who majored in psychology or economics or participated in any other drug study in recent weeks were excluded from participation. Participants were instructed to refrain from smoking or drinking (except water) for 2 hr before the experiment. The experiment involved no deception, and participants were paid for their presence for the experiment (i.e. $10 show-up fee) plus their average earnings in two randomly selected IADC rounds.

### Ethics approval

All participants provided written informed consent to participate after the experimental procedures had been fully explained, and were acknowledged their right to withdraw at any time during the study. All experimental procedures adhered to the standards set by the Declaration of Helsinki and were approved by the Institutional Review Board at the State Key Laboratory of Cognitive Neuroscience and Learning, Beijing Normal University, Beijing, China (protocol number: ICBIR_A_0107_001).

### Procedure

Participants were randomly assigned to the intranasal administration of oxytocin or placebo in a double-blind placebo-controlled between-subjects design (*Figure 1*). For each IADC session, six strangers were invited to the lab at the same time and randomly assigned to six individual cubicles within the same room. Upon arrival, participants first completed questionnaires that measured current mood, empathic capacity, prosocial personality, impulsiveness, subjective socio-economic status, and cooperative personality. Participants then self-administered oxytocin or placebo. After 35 min, participants were given instructions for the IADC game and completed two practice rounds. When they also passed a comprehension check, they played 15 simultaneous rounds and 15 sequential rounds of IADC investments (order of simultaneous and sequential blocks was counterbalanced across sessions, *Figure 2*). All the experimental instructions used neutral language (e.g. contribution was labeled investment; defense and attack were avoided, and groups were labeled as group A or B). Finally, participants filled out a post-survey for mood measurement and manipulation check. The attacker and defender groups under oxytocin or placebo did not differ in demographic information, mood change, and prosocial-related traits (*Supplementary file 1, Table 1A and B*).

## Oxytocin administration

The procedure of oxytocin and placebo administration was similar to previous work that showed oxytocin effects on decision-making behaviors or in-group favoritism (*De Dreu et al., 2010*; *Ma et al., 2015*; *Rilling et al., 2012*; *Yan et al., 2018*). A single intranasal dose of 24 IU oxytocin or placebo (containing the active ingredients except for the neuropeptide) was self-administered by nasal spray about 35 min before the experimental task under experimenter supervision. A 24 IU dosage is the most commonly used dosage in oxytocin literature (*Wang et al., 2017*) and recently shown as having more pronounced effects (compared with 12 or 48 IU dose of oxytocin) on behavioral and neural responses (*Spengler et al., 2017*). The spray was administered to participants three times, and each administration consisted of one inhalation of 4 IU into each nostril. Six participants in the same IADC session were assigned to the same treatment (oxytocin or placebo), so as to avoid potential influence of oxytocin to placebo between individuals.

## Data analysis

Data were aggregated to the group level, with Role (Attacker vs. Defender), Procedure (Simultaneous vs. Sequential) and Round as within-group variables, and Treatment (Oxytocin vs. Placebo) as a between-group factor. Analyses were performed on (i) contribution (the averaged investment of each round, range: 0–20), (ii) the number of non-contributors (the number of group members making a 0 contribution across a 15-round block, range: 0–45), (iii) variance (within-group variance of each round), (iv) success rate for attack (range 0–100%), (v) the attackers' leftovers when losing a conflict, (vi) the attackers' spoils when winning a contest and, finally, (vii) inter-group tracking. The within-group variance is calculated for each decision round, thus reveals the group coordination of contribution at each round, with lower variance indicating stronger coordination. In addition, to complement and validate these analyses, we analyzed (viii) decision time for investment decisions (log 10-transformed decision time), and (ix) within-group coordination as reflected in the Intra-class correlation coefficient.

## Game-theoretic analysis

The IADC is an all-pay contest with a single mixed-strategy Nash equilibrium (*Abbink et al., 2010*; *Dechenaux et al., 2015*). With two three-person groups, each member assumed to have risk-neutral preferences and having a discretionary resource of $e$ = 20 MU to invest from, the IADC game has unique mixed-strategy Nash equilibrium with out-group attack (in-group defense) expected to average 10.15 (9.77) across rounds, and attackers (defenders) should win (survive) 32.45% (67.55%) of the contest rounds (*De Dreu et al., 2016a*). Across the 15 rounds of simultaneous decision-making under placebo, both out-group attack and in-group defense fell below the Nash-equilibrium ($t(39)$ = −11.30, p < 0.001 and $t(39)$ = −4.18, p < 0.001). Attackers defeated defenders in 22.08% (SE = 2.2%) of their attacks, which is below the Nash success-rate ($t(39)$ = −4.734, p < 0.001). Oxytocin did not influence deviations from rationality (attack: $t(39)$ = −15.78, p<0.001; defense: $t(39)$ = −6.022, p<0.001; success-rate: $t(39)$ = −6.615, p < 0.001).

## Response time for decisions to (not) contribute

We showed that oxytocin increases the number of non-contributors in attacker groups. To further reveal how oxytocin influenced the non-contributing decisions, we examined participants' decision time by calculating the response time separately for the round that participants decided to or not to contribute. This analysis was conducted only on the attacker group because (1) there were very few rounds (9%) in which defenders decided not to contribute; (2) oxytocin selectively influenced the number of non-contributors in attacker groups. Since the distribution of decision times is heavily right skewed, linear regression is not appropriate. Similar to our previous study (*Ma et al., 2015*), we first log 10-transformed decision times in all the analyses that involved decision times. The log 10-transformed decision time was submitted into a 2 (Contribute: Yes vs. No) × 2 (Treatment: Oxytocin vs. Placebo) ANOVA for the attacker group. The decisions of which the response time exceeded 180 seconds were excluded in final data analysis (0.51% of the decisions, due to network problem during the experiment).

## Within-Group coordination

To complement and cross-validate the results for within-group variance as an indicator of within-group coordination, we computed another related index — the intra-class correlation. The intraclass correlation (ICC, *De Dreu et al., 2016a*; *LeBreton and Senter, 2008*) operates on a data structured as groups, rather than data structured as paired observations. ICC describes the amount of statistical interdependence within a group (group cohesion), reflects how strongly individuals' contributions in all rounds in the same group resemble each other, that is how similar group members are in their contributions to the group pool across rounds. Higher ICC values in essence mean group members are more similar to each other in the contributions made to their group pool. A Treatment $\times$ Role $\times$ Procedure ANOVA showed effects for Role ($F$(1,78) = 43.090, p < 0.001, $\eta^2$= 0.356), Procedure ($F$(1,78) = 166.199, p < 0.001, $\eta^2$ = 0.681) and for the Role x Procedure interaction (F(1,78) = 147.586, p < 0.001, $\eta^2$ = 0.654). Fitting the results for within-group variance reported in the Main Text (*Figure 5A*), results further showed that oxytocin increased attacker groups' ICC under simultaneous decision-making (t(78) = 2.057, p = 0.043, Cohen's d = 0.460, *Figure 5—figure supplement 1*; not under sequential block: t(78) = -0.179, p = 0.859, Cohen's d = 0.040), but did not influence defender groups' ICC (simultaneous: t(78) = 0.485, p = 0.629, Cohen's d = 0.108; sequential: t(78) = 0.389, p = 0.698, Cohen's d = 0.087).

## Tracking on historical contribution

To test whether attacker groups made their contributions based on tracking of their rival's historical level of defense, we built a multiple linear regression of attacker groups' average contribution on round j (referred as $A_j$, with j range from 3 to 15) as a function of average level of defense of last rounds (calculated as ($D_{j-1}$ +$D_{j-2}$)/2, regression weight referred as $\alpha$). The regression weight $\alpha$ was Fisher's z transformed for statistical analysis. Similar to a previous study (*De Dreu et al., 2016b*), we also included another parameter: defense change of the last and before-last rounds, calculated as ($D_{j-1}$ - $D_{j-2}$), regression weight referred as $\beta$. However, the analysis on $\beta$ failed to show significant main effects of Treatment/Procedure or their interaction (*ps* >0.1). To complement the analysis of attacker groups, we also examined whether and how defender groups tracked the historical level of attack in their rivals. This showed that defender groups relied more on $\alpha$ to track attacker groups under simultaneous (relative to sequential) decision-making. The main effect of Treatment and its interaction with Procedure were not significant (ps >0.05).

*Mediation analysis.* We performed formal mediation analyses to examine through which route oxytocin increased attacker group's spoils from winning a conflict. Two potential mediators were included in the mediation model: one is tracking coefficient ($\alpha$) the other is within-group variance. Four different regression models were constructed, as shown below:

$$Y = \beta_{11}X + \beta_{10} \tag{1}$$

$$M1 - \beta_{21}X + \beta_{20} \tag{2}$$

$$M1 = \beta_{31}X + \beta_{32}MI + B_{30} \tag{3}$$

$$Y = \beta_{41}X + \beta_{42}MI + B_{43}M2 + B_{40} \tag{4}$$

In these models, X is the independent variable (Treatment, dummy-coded, 0 for placebo and one for oxytocin), M1 is the first mediator (the weight for attacker group's tracking of the historical defense, the tracking coefficient, $\alpha$), M2 is a second mediator (attacker group's within-group variance), and Y is the dependent variable (attacker groups' spoils from winning a conflict, reported in the Main Text, and DV with the sum of attacker groups' spoils from winning a conflict and leftovers from losing a conflict reported in the *SI*). A resampling method known as bootstrapping was used to test the direct and indirect path. Bootstrapping is a nonparametric approach to effect-size estimation and hypothesis testing that is increasingly recommended for many types of analyses, including mediation (*Mackinnon et al., 2004*; *Shrout and Bolger, 2002*). Rather than imposing questionable distributional assumptions, bootstrapping generates an empirical approximation of the sampling distribution of a statistic by repeated random resampling from the available data, and uses this

distribution to calculate p-values and construct confidence intervals (5000 resamples were taken for these analyses). Moreover, this procedure supplies superior CIs that are bias-corrected and accelerated (*Preacher et al., 2007*). Results are summarized in *Figure 6D*, *Figure 6—figure supplement 1*, *Supplementary file 1, Table 1C and D*. As can be seen, multistep mediational analysis showed that the oxytocin effect on increasing attacker group's spoils from winning the conflict *plus* leftovers from losing the conflict was mediated by its effect on increasing tracking of defenders' history so as to increase within-group coordination.

## Acknowledgements

This work was supported by the National Natural Science Foundation of China (Projects 31722026; 31771204; 91632118; 31661143039 to YM); the Fundamental Research Funds for the Central Universities (2016NT05; 2017XTCX04; 2018EYT04 to YM); Beijing Municipal Science and Technology Commission (Z151100003915122 to YM); Open Research Fund of the State Key Laboratory of Cognitive Neuroscience, Beijing Normal University; startup funding from the State Key Laboratory of Cognitive Neuroscience and Learning, IDG/McGovern Institute for Brain Research, Beijing Normal University to YM; and the Spinoza Award from the Netherlands Science Foundation (NWO SPI-57–242 to CKWDD). We thank Shiyi Li for assistant in data collection and the members of the SANP lab at Beijing Normal University, and the members of the social-decision making laboratory at Leiden University for comments on an earlier draft.

## Additional information

### Funding

| Funder | Grant reference number | Author |
|---|---|---|
| National Natural Science Foundation of China | 31722026 | Yina Ma |
| National Natural Science Foundation of China | 31771204 | Yina Ma |
| National Natural Science Foundation of China | 31661143039 | Yina Ma |
| National Natural Science Foundation of China | 91632118 | Yina Ma |
| Fundamental Research Funds for the Central Universities | 2018EYT04 | Yina Ma |
| Fundamental Research Funds for the Central Universities | 2017XTCX04 | Yina Ma |
| Spinoza Award from the Netherlands Science Foundation | NWO SPI-57-242 | Carsten De Dreu |
| Beijing Municipal Science and Technology Commission | Z151100003915122 | Yina Ma |

The funders had no role in study design, data collection and interpretation, or the decision to submit the work for publication.

### Author contributions

Hejing Zhang, Software, Formal analysis, Investigation, Visualization, Methodology; Jörg Gross, Software, Methodology; Carsten De Dreu, Conceptualization, Visualization, Methodology, Writing—original draft, Writing—review and editing; Yina Ma, Conceptualization, Resources, Formal analysis, Supervision, Funding acquisition, Visualization, Methodology, Writing—original draft, Writing—review and editing

## Author ORCIDs
Jörg Gross (iD) https://orcid.org/0000-0002-5403-9475
Carsten De Dreu (iD) http://orcid.org/0000-0003-3692-4611
Yina Ma (iD) http://orcid.org/0000-0002-5457-0354

## Ethics
Human subjects: All participants provided written informed consent to participate after the experimental procedures had been fully explained, and were acknowledged their right to withdraw at any time during the study. All experimental procedures adhered to the standards set by the Declaration of Helsinki and were approved by the Institutional Review Board at the State Key Laboratory of Cognitive Neuroscience and Learning, Beijing Normal University, Beijing, China (protocol number: ICBIR_A_0107_001).

## Decision letter and Author response
Decision letter https://doi.org/10.7554/eLife.40698.017
Author response https://doi.org/10.7554/eLife.40698.018

# Additional files

### Supplementary files
• Supplementary file 1. Supplementary Table 1. (A) Table 1A. Match demographic information and prosocial-related traits. (B) Table 1B. Mood changes from pre-experiment to post-experiment. (C) Table 1C. Point estimates for indirect effects and bootstrapped 95% bias-corrected confidence intervals for multiple mediational analysis in which attacker group's tracking (*strategic tracking when* $\alpha \to -1$) and within-group variance (*variance*) were represented as mediators in the association between *Treatment* and *spoils from winning* a conflict during simultaneous decision-making. (D) Table 1D. Point estimates for indirect effects and bootstrapped 95% bias-corrected confidence intervals for multiple mediational analysis in which attacker group's tracking (*strategic tracking when* $\alpha \to -1$) and within-group variance (*variance*) were represented as mediators in the association between *Treatment* and *spoils and leftovers* during simultaneous decision-making.
DOI: https://doi.org/10.7554/eLife.40698.012
• Transparent reporting form
DOI: https://doi.org/10.7554/eLife.40698.013

### Data availability
The group-level statistics are plotted in Figures 3-6, Figure 5-figure supplement 1 and Figure 6-figure supplement 1.The individual data is plotted in Figures 6 and Figure 6-figure supplement 1. The source data for Figures 3-6 is publicly available on the Open Science Framework at https://osf.io/7jeas/ (doi: 10.17605/OSF.IO/7JEAS).

The following dataset was generated:

| Author(s) | Year | Dataset title | Dataset URL | Database and Identifier |
|---|---|---|---|---|
| SANP_Neuroscience, Hejing Zhang, Yina Ma | 2018 | Oxytocin promotes coordinated out-group attack during intergroup conflict in humans | https://osf.io/7jeas/ | Open Science Framework, 10.17605/OSF.IO/7JEAS |

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
