## [Decision Letter]

[**Editorial note:** This article has been through an editorial process in which the authors decide how to respond to the issues raised during peer review. The Reviewing Editor's assessment is that all the issues have been addressed.]

Thank you for submitting your article "Oxytocin promotes synchronized out-group attack during intergroup conflict in humans" for consideration by *eLife*. Your article has been reviewed by four peer reviewers, including Peggy Mason as the Reviewing Editor and Reviewer #1, and the evaluation has been overseen by Timothy Behrens as the Senior Editor. The following individuals involved in review of your submission have also agreed to reveal their identity: Caroline F Zink (Reviewer #2); Takefumi Kikusui (Reviewer #3).

The Reviewing Editor has highlighted the concerns that require revision and/or responses, and we have included the separate reviews below for your consideration. If you have any questions, please do not hesitate to contact us.

The reviewers agreed on the value of the data generated in this study. The sample size is large and well controlled. There were concerns around clarity of methods and results that the authors may want to consider to most efficaciously reach the widest audience. There were also two primary areas where the reviewers would urge the authors to either adjust their interpretation or at least acknowledge the possibility of other interpretations.

Playing a monetary computer game is not dancing together to go to war. The behavioral–motor synchrony is simply not at play here. This analogy should be dialed back.

More fundamental to the point the authors purport to make are simpler explanations for the results than attack coordination. For example, it could be "that a primary role of oxytocin [is] to increase one's own (at an individual level) social intuition and acuity in the assessment of how strong or challenging a defense may be and when and how it would be effective to attack as a result (along the lines of social adaptation)" (Reviewer #2) or that "oxytocin increased the participants' memory for the sequence of events (there is evidence for oxytocin effects on memory) or that it reduced their level of stress and increased calmness and lethargy (there evidence in both humans and animals) and they reduced their contributions" (Reviewer #4).

We look forward to seeing this exciting contribution to our understanding of OXT and its effects on behavior. Your findings are more nuanced than the extreme version favored by the public and curiously still adopted by many scientists even beyond those heavily invested in the OXT-as-positive social modulator view. Thank you for communicating this work to us.

Separate reviews (please respond to each point):

*Reviewer #1:*

Overall this is an exciting contribution to our understanding of OXT and its effects on behavior with a result that is more nuanced than the extreme version favored by the public and curiously still adopted by many scientists even beyond those heavily invested in the OXT-as-positive social modulator view.

Beyond the rhetorical issues detailed in Minor comments, my major concern is that the authors use motor and social terms for what is an in silico experiment. Participants never did anything active beyond pressing a computer key. To compare in silico coordination to a "war dance" misses the criticality of movement and motor mimicry, which do not play a role here. It is in my opinion even a bit of hyperbole or slight of hand to rely so heavily on the motor-laden words of attack and defend. This is really some economic version of hostile takeover/ investment. The actions have no inherent meaning and the motor system does not come directly into play. I would ask that the authors tone down the motoric rhetoric.

Minor Comments:

Evolutionary ancient is a poor term (what does it even mean?)- I believe that phylogenetically widespread or conserved would be a more scholarly way to put it.

The distribution of intranasal OXT is controversial and the statement in the second paragraph of the Introduction should be toned down. Most of the refs cited have nothing to do with distribution (eg Burkett is on effects of OXT on prairie vole behavior) and for example the Paloyelis paper discusses changes in blood flow that may be direct but just as easily could be the indirect result of OXT actions. That the jury is out regarding where OXT works with what time course is okay to acknowledge. Please don't overstate the case.

Figure 1 typo spray NOT spay

Figure 2 is confusing. in the simultaneous block, why would the attackers get 17 when they lost 7 to 6? By your description, Results paragraph one, they should hold onto 14 rather than "earn 17." Then in the sequential rounds, why is the last one 18 instead of 16? And really – why would you choose a fraction? When the attackers win 10-7, the win (spoils) is (60-7)/3 = 17.8 rounded down to 17 and that is before adding to their remaining MU? But instead it is 16, 16, and 18. Spell it out please or fix the description in the Materials and methods. Something is not right.

Why use both correlation and variance? And why not just use SD?

Results paragraph six: typo rival s'. --- rivals'

Explain the bootstrapping. This is a general audience. Make it understandable to the general scientific reader.

Materials and methods subsection “Procedure”; Several of the mood measures did change (Supplementary file 1) and in a highly interesting way. Why is it stated that none changed?

*Reviewer #2:*

This fascinating investigation of oxytocin (OT) in intergroup conflict is interesting, important, and timely. It is creative and innovative with regards to the paradigm, analyses, and interpretations. In addition to innovation, additional strengths include a large sample size for adequate power, and breadth of analyses performed. I did notice that some of the most important statistical tests for the conclusions were not significant, but rather statistically trending (Results section) which in my opinion should be noted/discussed as such. I am curious how the subjects were paid. Was their final payment associated with their performance in the task? What is the interpretation of OT increasing the number of non-contributors during attack? The data supporting the notion that OT does not increase contributions to attack in general, but rather coordinates attacking at the right moment with the right force is quite compelling, especially being more pronounced in the simultaneous rather than sequential procedure. To make that point explicitly in terms of oxytocin increasing payment (leftovers and spoils) more during the simultaneous procedure, it would be nice to see the analyses pertaining to that broken down by procedure (i.e. simultaneous and sequential separately). Without debating semantics, which is not my intention, I am also struck by the interpretation that OT is a biological mechanism to converge or "coordinate" contributions between individuals in the attack group (when not sequential). It seems to me that the concept of coordination or convergence is secondary to perhaps a primary role of oxytocin to increase one's own (at an individual level) social intuition and acuity in the assessment of how strong or challenging a defense may be and when and how it would be effective to attack as a result (along the lines of social adaptation). In the simultaneous context, it would be evolutionarily important for each individual to act at maximum efficiency because there is no leader or others to rely on as there is in the sequential procedure. So by OT increasing each individuals' social acuity and using that information to act most adaptively or effectively, as a group they will appear coordinated (acting similarly) in their beneficial action and together will achieve success. In this case, OT would not be facilitating the social coordination per se, but rather individual social acuity and reaction assessment that is necessary to achieve success as a group in the simultaneous context. Also, I am not convinced that this study demonstrated that social bonding and performing synchronization rituals helps coordinate attacks (as put forth in paragraph two of the Discussion) because as I understand it, in this study group members did not interact or socially bond before coordinating their attacks. Overall, I was extremely impressed with the caliber of this thought-provoking manuscript and believe that is positively contributes to the field and scientific community.

Minor Comments:

Many of the figures legends are lacking definitions of the symbols used to denote significance (i.e. what significance level). Also the key is missing in Figure 5—figure supplement 1.

Discussion, third paragraph final sentence: should it be *not* needed?

Discussion paragraph four: "remains" should be *remained* past tense because it does not remain with the current study.

*Reviewer #3:*

This paper demonstrated that in an inter-group conflict game, exogenous oxytocin (OT) treatment increased attacker's within-group synchronization of contributions (reduced within-group variance). These behavioral changes would be because of selecting more effective strategy (attacking at a proper time when the defender was weak). This within-group coordination in attacking can increase spoils from winning. The aim of this study was based on their previous studies and was clear, and the methods were well-conducted. The number of the participants were well-enough to draw the conclusions. However, I have some concerns/questions on this form of the manuscript.

1) In Figure 1A, the authors mentioned that contribution amount was low in the subjects in the OT Condition. However, the authors already reported that OT promotes internal group defense (as a review De Dreu and Kret 2016 https://www.ncbi.nlm.nih.gov/pubmed/25908497). Therefore, it is necessary to present the data comparing Contribution amounts among four levels (Treatment × Role). If there was inconsistency between their previous study and present results regarding to the effects of OT on contribution in defense group, the authors need to discuss this.

2) In Figure 1B, the data were inconsistent with De Dreu et al., 2016, in that there is no difference in the frequency of Non-contributors between Simultaneous and Sequential. The authors need to explain the inconsistency.

3) In the same figure, in the Attack group of the Simultaneous and OT condition, did the frequency of Non-contributors increase in rounds in which attacks fail, and did the frequency of Non-contributors decrease when the attack succeeded? The information on the frequency of non-contributors when dividing by attacking success / failure is necessary.

4) If OT changes the attack efficiency, it is thought that information on how the defense parameter α has a negative regression coefficient to influence the success rate of attack is necessary.

5) In Discussion, the authors discussed one possibility that in-group defense was tacitly well-coordinated because of the stronger alignment of individual interests within defender groups. If this is true, would the results change by modulating the ratio of losing/gaining MU as the results of attach/defense? How did the authors decide the ratio of cost/benefit of the MU after the challenge?

*Reviewer #4:*

In this study, the authors examined the effects of oxytocin administration on a large sample of Chinese men (n = 480) engaged in a three-person monitory game divided to "attackers" and "defenders" where the goal was to evaluate donation to the group monetary pool in relation to the donation of own group and opponent group.

It was found that oxytocin reduced donation to the group pool in the "attacker" group and increased the "coordination" between the participants' donation, defined as the inter-subject correlation between the parents' contribution. It also increased the "attackers" ability to track the opponents' contribution in sequential trials.

The large sample size is an asset of the study. It is, to my knowledge, the largest oxytocin administration study conducted so far. The game is complex and so are the statistical analyses.

The major issue with the study, in my opinion, is whether it measures what it claims to show. It is unclear whether the economical game the authors developed can provide any simulation to real-life battle, "warrior games" or "coordinated action" (the theoretical frame they set for the study), whether inter-subject correlations between monetary contributions of individuals and reduced variance indeed show "group synchrony" in the face of external danger, whether survival has anything to do with the neurobiological processes that underpin this game, and whether there are alternative explanations to the findings that are more plausible and parsimonious (and more consistent with findings from studies on oxytocin administration and economical games).

In terms of "coordinated action" or "behavioral synchrony" among groups, the increased group cohesion in the face of danger, and the rites and rituals group perform before going to battle, these are indeed very powerful mechanisms and are found across human cultures and non-human primates. What is unclear is how coordinated action patterns relate to contributing more or less to the general pool in a sequential economic game. There are ways to measure action coordination in groups, in terms of real action patterns, mimicry, imitation, facial expressions, coordinated movement etc, and the increase in coordination results from various factors in the group, for instance, the presence of a charismatic leader (Gordon, I. Berson, Y. Oxytocin modulates charismatic influence in groups. J Exp Psychol Gen. 2018 Jan;147(1):132-138. doi: 10.1037/xge0000375). It is unclear at all why the authors measure "behavioral synchrony" as the reduced variance among the contributions of members and it is not possible, in my opinion, to simply link this reduced variable to the literature on behavioral synchrony, action coordination, and war rituals without some kind of validation that this reduced variance in the differences of contribution over trials is indeed related to more behavioral coordination. Without such indication, it is not possible to propose that oxytocin, which is indeed a mechanism to enhance coordination of behavior (movements, vocalizations, mimicry) in humans and non-human primates, to the present findings.

An additional difficulty is the claim that literature on survival is relevant to this game in any way and that playing this game simulates mechanisms of survival. Here, I was unclear about the difference between "attackers" and "defenders" on a biological level. It appears that one group was told that they "attack" something and another that they "defend" something but attack and defend where not part of the ongoing game, even not at the level of a simple board game where there is a simulation of "defending a castle", but that the real game, for the two groups, was identical was about how to outsmart the other group in terms of donating more or less. The distance between this and proposing that the two groups differ on biological mechanisms is substantial, and must be demonstrated prior to conducting this study. It is more parsimonious (and plausible) to conclude that the mechanisms underpinning this game related more to human competition and are supported by humans' large prefrontal abilities, working memory, and far-sighted planning, and are unrelated to "movement coordination" in the face of a threat to survival. Such mechanisms are likely to be identical for the attackers and defenders (unless otherwise proven) and probably have much less to do with more primitive mechanisms described in the animal literature. For instance, the primate study the authors cite (Summani) showed greater vocal coordination and greater urinary oxytocin in primates during live conflict and is too great a leap to hypothesize from this to reduced variance of donation among players in a safe room. This study also did not differentiate between the behavioral repertoire (and oxytocin production) of attack (where motion and vocal synchrony correlated with oxytocin) and those related to "border patrol", which suggests that in the animal kingdom, both attack and defense are part of the survival and may not be biologically distinct.

The inter-subject correlation – the authors' index for behavioral synchrony – was measured only in the "attacker" group (rationale not clear) but this group also markedly reduced its contribution over time under oxytocin. It is possible that the increase in correlation among group members resulted from that decrease and that the zero donation correlated among the participants (even if we accept the suggestion that this is a proxy for synchrony, it is a synchrony of "no action" rather than of action).

A more parsimonious explanation for the findings – and one that must be ruled out before offering a mechanistic explanation based on rituals and action coordination – would be that oxytocin increased the participants' memory for the sequence of events (there is evidence for oxytocin effects on memory) or that it reduced their level of stress and increased calmness and lethargy (there evidence in both humans and animals) and they reduced their contributions. Even within the literature on oxytocin and economic games (which should provide the conceptual frame for this study) the findings need to be reconciled with those showing that individuals donate more under oxytocin. Is this a cultural phenomenon specific to the Chinese culture?

Another important issue is the between-subject design, which is not the golden standard in oxytocin administration research. Particularly in relation to economic games, a recent study showed different brain activation patterns in a within- and between-subject design in an OT administration study using the prisoner's dilemma (Chen X, Gautam P, Haroon E, Rilling JK Within vs. between-subject effects of intranasal oxytocin on the neural response to cooperative and non-cooperative social interactions. Psychoneuroendocrinology. 2017 78:22-30. doi: 10.1016/j.psyneuen. 2017.01.006). At least, the authors should describe how randomization was conducted and show extensive comparisons among groups not only in demographics but also in relevant physiological variables (for instance, BMI).

In sum, the authors have collected an impressive sample and the study is complex and well-analyzed. The conceptual frame should fit the study design and alternative explanation must be carefully considered before contribution of the findings can be considered. As frames, the conclusions of the study are not supported by the data and design and may be misleading.

Additional data files and statistical comments:

No need for more data. The study does not measure behavioral synchrony under conflict and there is a great gap between measuring the degree of sameness in how much individuals contributed (or didn’t contribute) and the survival evolutionary frame, war rituals, and action coordination the authors invoke to support the findings. In the current frame and design, the conclusions are misleading.

---

## [Author Response]

The reviewers agreed on the value of the data generated in this study. The sample size is large and well controlled. There were concerns around clarity of methods and results that the authors may want to consider to most efficaciously reach the widest audience. There were also two primary areas where the reviewers would urge the authors to either adjust their interpretation or at least acknowledge the possibility of other interpretations.Playing a monetary computer game is not dancing together to go to war. The behavioral – motor synchrony is simply not at play here. This analogy should be dialed back.More fundamental to the point the authors purport to make are simpler explanations for the results than attack coordination. For example, it could be "that a primary role of oxytocin [is] to increase one's own (at an individual level) social intuition and acuity in the assessment of how strong or challenging a defense may be and when and how it would be effective to attack as a result (along the lines of social adaptation)" (Reviewer #2) or that "oxytocin increased the participants' memory for the sequence of events (there is evidence for oxytocin effects on memory) or that it reduced their level of stress and increased calmness and lethargy (there evidence in both humans and animals) and they reduced their contributions" (Reviewer #4).

We appreciate these helpful suggestions. Accordingly, our revision stays closer to the data and removes any language that refers to behavioral-motor synchrony. For example, the reference to behavioral synchronization was replaced by coordination of investment or coordinated contribution; intergroup conflict was replaced by intergroup competition and conflict. We also noted and/or discussed other possibilities to interpret the current finding, as suggested by the reviewers #2 and #4 in the revised manuscript.

Reviewer #1:

Overall this is an exciting contribution to our understanding of OXT and its effects on behavior with a result that is more nuanced than the extreme version favored by the public and curiously still adopted by many scientists even beyond those heavily invested in the OXT-as-positive social modulator view.Beyond the rhetorical issues detailed in Minor comments, my major concern is that the authors use motor and social terms for what is an in silico experiment. Participants never did anything active beyond pressing a computer key. To compare in silico coordination to a "war dance" misses the criticality of movement and motor mimicry, which do not play a role here. It is in my opinion even a bit of hyperbole or slight of hand to rely so heavily on the motor-laden words of attack and defend. This is really some economic version of hostile takeover/ investment. The actions have no inherent meaning and the motor system does not come directly into play. I would ask that the authors tone down the motoric rhetoric.

We’re grateful to the reviewer for this helpful suggestion. We agree with the reviewer and take care of our language to avoid the link to movement or motor action. For example, we omitted the term behavioral synchronization and deleted the ‘war dance’ metaphor. We did keep, however, the terms ‘attack/attacker’ and ‘defense/defender’ in the revision for two reasons. First, we use these terms to keep consistency with the literature as these terms were used in the original economic game (De Dreu et al., 2016; in press). Second, we used the terms ‘attack/attacker’ and ‘defense/defender’ to indicate that the two parties in the current economic game are in different positions (an asymmetric structure of attacker and defender), with one side aiming to increase gains through victory (but always keep the remaining resources), and the other side aiming to protect against loss and defeat (i.e., they keep their remaining resources only when defense succeed; otherwise they are left with 0).

In the revision, we explicitly stated that the terms ‘attack/attacker’ and ‘defense/defender’ were used in the context of economic games to avoid the link to movement or motor action. Moreover, we included specific definitions of out-group attack and in-group defense (Introduction and Results section).

Minor Comments:Evolutionary ancient is a poor term (what does it even mean?)- I believe that phylogenetically widespread or conserved would be a more scholarly way to put it.

We thank reviewer #1 for pointing this out and we revised accordingly throughout the manuscript.

The distribution of intranasal OXT is controversial and the statement in the second paragraph of the Introduction should be toned down. Most of the refs cited have nothing to do with distribution (eg Burkett is on effects of OXT on prairie vole behavior) and for example the Paloyelis paper discusses changes in blood flow that may be direct but just as easily could be the indirect result of OXT actions. That the jury is out regarding where OXT works with what time course is okay to acknowledge. Please don't overstate the case.

We do agree with reviewer #1 that the pathway intranasal oxytocin takes to the central system is unclear and controversial. In the revised manuscript we have tried to be more nuanced. For example, we now refer to studies showing evidence that intranasal oxytocin modulates neural responses in brain regions involved in threat detection and reward processing without making claims about the mediating neurophysiological pathways (Introduction paragraph two).

Figure 1 typo spray NOT spay

We thank reviewer #1 for pointing this out, and we have accordingly revised Figure 1.

Figure 2 is confusing. in the simultaneous block, why would the attackers get 17 when they lost 7 to 6? By your description, Results paragraph one, they should hold onto 14 rather than "earn 17." Then in the sequential rounds, why is the last one 18 instead of 16? And really – why would you choose a fraction? When the attackers win 10-7, the win (spoils) is (60-7)/3 = 17.8 rounded down to 17 and that is before adding to their remaining MU? But instead it is 16, 16, and 18. Spell it out please or fix the description in the Materials and methods. Something is not right.

We apologize for not being clear and revised Figure 2 accordingly. Moreover, we added a table to provide a clear description of the rule for the IADC game, as well as how the payment is calculated for each individual (i.e., Table 1 in the revision). Regarding the specific examples in Figure 2: in the simultaneous block, we illustrated one round for an attacker group who lose this round (investing 6 in total vs. a defender group’s pool of 7) leaving each attacker with what was left from their contribution: Attacker 1 earned 17 (as 20-3), Attacker 2 earned 19 (as 20-1), and Attacker 3 earned 18 (as 20-2). In the sequential block, we illustrated one round for a defender group who win this round (investing 10 in total vs. an attacker group’s pool of 7) thus leaving each defender with what was left from their contribution: Defender1 earned 16 (as 20-4), Defender 2 earned 16 (as 20-4), and Defender 3 earned 18 (as 20-2).

Why use both correlation and variance? And why not just use SD?

We thank reviewer #1 for raising these questions, thus giving us the opportunity to clarify these measures in the current study (Results paragraph two; Materials and methods subsections “Data analysis” and “Within-Group Coordination”). We used the correlation coefficient (intraclass correlation coefficient, ICC) and variance to indicate different aspects of the group-level coordination (contribution similarity of members of the same group), with the correlation coefficient reflecting group decision similarity across all rounds and variance reflecting the differences between each group member and the group mean on a round-by-round basis.

– Regarding the use of variance and ICC:

The ICC (LeBreton and Senter, 2008) operates on a data structured as groups, rather than data structured as paired observations. ICC describes the amount of statistical interdependence within a group (group cohesion), reflects how strongly individuals’ contributions in all rounds in the same group resemble each other, i.e., how similar group members are in their contributions to the group pool across rounds. Higher ICC values in essence mean group members are more similar to each other in the contributions made to their group pool. On the other hand, the within-group variance is calculated for each decision round, thus reveals the group coordination of contribution for each round, with lower variance indicating stronger coordination. Thus we can examine the effects of Role, Procedure and Treatment on group variance, as well as to reveal how these effects on group coordination change across rounds.

For both indices, we observed a similar pattern that oxytocin increased within-group coordination in attackers especially when decisions were made simultaneously. Thus, we reported the effect on within-group variance in the main text, and reported the similar effect on ICC in the SI as a further support.

– Regarding the choice of variance instead of SD:

The standard deviation can be calculated out of variance, as the square root of the variance. The results would be the same. We chose to use variance to keep consistency with the literature (De Dreu et al., 2016; 2019).

Results paragraph six: typo rival s'. --- rivals'

Thanks and corrected accordingly.

Explain the bootstrapping. This is a general audience. Make it understandable to the general scientific reader.

We thank the reviewer for this suggestion, accordingly we provided a clearer description of how the bootstrapping is done (Results paragraph six).

Materials and methods subsection “Procedure”; Several of the mood measures did change (Supplementary file 1) and in a highly interesting way. Why is it stated that none changed?

We checked the mood measures in Supplementary file 1, Supplementary Table 1B again but confirmed that no Treatment x Role interactions on mood change was found (not for the positive and negative mood separately, nor for the overall mood change). We realize that the presentation of both F and p values might be confusing to the audience, thus we only included the Mean (SE) and p-values in the revised Supplementary file 1. Supplementary Tables 1B. We conducted the paired t-test (before- vs. after-experiment mood) for each group to examine mood change for each group. None of the group showed significant change in positive mood or negative mood (ps > 0.05). Moreover, ANOVAs with Treatment and Role as between-subjects factors on the positive, negative, and overall mood changes also did not show significant main effects of Treatment and Role, or Treatment x Role interaction (ps > 0.05).

Reviewer #2:

This fascinating investigation of oxytocin (OT) in intergroup conflict is interesting, important, and timely. It is creative and innovative with regards to the paradigm, analyses, and interpretations. In addition to innovation, additional strengths include a large sample size for adequate power, and breadth of analyses performed.

We appreciate the positive and enthusiastic evaluation and thoughtful comments from the reviewer.

I did notice that some of the most important statistical tests for the conclusions were not significant, but rather statistically trending (Results section) which in my opinion should be noted/discussed as such.

We agree with reviewer #2 and now explicitly noted and/or discussed these marginally significant results in the revision.

I am curious how the subjects were paid. Was their final payment associated with their performance in the task?

The experiment involved no deception, and participants were paid for their presence for the experiment (i.e., $10 show-up fee) plus their average earnings in two randomly selected IADC rounds. This information was clarified in the revision (Materials and methods subsection “Participants”).

What is the interpretation of OT increasing the number of non-contributors during attack?

We thank the reviewer for this question. There were two potential interpretations for the effect of oxytocin on increasing non-contributors during attack. On the one hand, oxytocin has been linked to prosociality, empathy and consolation (Rilling et al., 2012; Yan et al., 2018; Hurlemann et al., 2010; Bartz et al., 2010; Burkett et al., 2016). This line of research would suggest that oxytocin enhances attackers’ prosocial behavior, including a desire for “peaceful” no-attack (i.e., not contributing). Alternatively, oxytocin has been shown to enhance in-group favoritism and conformity (Aydogan et al., 2017; Stallen et al., 2012; De Dreu and Kret, 2016), to increase behavioral coordination and neural synchronization (Arueti et al., 2013; Mu et al., 2016), and to improve memory and learning from feedback (Striepens et al., 2012; Adam et al., 2008; Ma et al., 2016). This line of research would suggest that oxytocin enables attackers to coordinate better on the level and timing of their attacks, and to efficiently appropriate resources from out-group rivals. Here non-contributions reflect a decision to “not attack now,” for example because the defenders appear strong and tough to win from.

As we discuss in the manuscript, if oxytocin enable groups to coordinate on a peaceful “no-attack strategy”, we would expect 1) lower victories in attacker groups given oxytocin than placebo, 2) the oxytocin effect on the number of non-contributing attackers should not differ when attacks succeed or fail, and 3) no effect of oxytocin on tracking rival’s defense history. Results from our further analyses on the success rate (oxytocin did not decrease success rate), non-contributor in successful or failed attacks (oxytocin increased non-contributing attacker only in failed but not in successful attacks), and tracking parameter (oxytocin increases attacker’s tracking of rival’s history, less attack when rivals defended strongly) lend supports for the second interpretation that oxytocin increases effective attack strategy rather than peaceful no-attack strategy. Consistent with previous studies, oxytocin may increase non-contributing attacker through its function in facilitating group-serving bias (De Dreu and Kret, 2016; De Dreu, et al., 2011), promoting social learning and cognitive flexibility (Ma et al., 2016a; Sala, et al., 2011), and increasing salience of social feedback (Shamay-Tsoory, et al., 2016).

The data supporting the notion that OT does not increase contributions to attack in general, but rather coordinates attacking at the right moment with the right force is quite compelling, especially being more pronounced in the simultaneous rather than sequential procedure. To make that point explicitly in terms of oxytocin increasing payment (leftovers and spoils) more during the simultaneous procedure, it would be nice to see the analyses pertaining to that broken down by procedure (i.e. simultaneous and sequential separately).

We thank the reviewer for this question. First, we would like to clarify that oxytocin enables coordinated attack at the right time with the right force is supported by the result of oxytocin effects on the tracking parameter. Oxytocin increased tracking on rival’s past defense, i.e., attackers given oxytocin attacked especially when defenders appeared more rather than less vulnerable. Then, following the reviewer’s suggestion, we broke down the Treatment x Procedure interaction on tracking parameter α by procedure to show the oxytocin effect on the tracking parameters in simultaneous and sequential decision-making separately. We showed that oxytocin increased attacker’s strategic tracking (i.e., a negative α) on rival’s history when decisions were made simultaneously (oxytocin vs. placebo: -0.30 ± 0.05 vs. -0.042 ± 0.098, t(78) = -2.334; p = 0.022, Cohen’s d = -0.522, 95% CI, -0.482 to -0.038), however, oxytocin decreased strategic tracking when decisions were made sequentially (oxytocin vs. placebo: -0.06 ± 0.05 vs. -0.18 ± 0.04, t(78) = 2.031; p = 0.046, Cohen’s d = 0.454, 95% CI, 0.003 to 0.249). This may reflect the fact that in sequential decision-making, people rely more on each other’s investments to coordinate, than on their rival group’s past behavior.

Without debating semantics, which is not my intention, I am also struck by the interpretation that OT is a biological mechanism to converge or "coordinate" contributions between individuals in the attack group (when not sequential). It seems to me that the concept of coordination or convergence is secondary to perhaps a primary role of oxytocin to increase one's own (at an individual level) social intuition and acuity in the assessment of how strong or challenging a defense may be and when and how it would be effective to attack as a result (along the lines of social adaptation). In the simultaneous context, it would be evolutionarily important for each individual to act at maximum efficiency because there is no leader or others to rely on as there is in the sequential procedure. So by OT increasing each individuals' social acuity and using that information to act most adaptively or effectively, as a group they will appear coordinated (acting similarly) in their beneficial action and together will achieve success. In this case, OT would not be facilitating the social coordination per se, but rather individual social acuity and reaction assessment that is necessary to achieve success as a group in the simultaneous context.

We thank the reviewer for this observation, and we agree that it is difficult to disentangle whether 1) oxytocin directly increases coordinated attacks or 2) oxytocin primarily increased individual-level performance, which results in seemingly coordinated group behavior. One key argument is that in our economic contest, *individual attackers* are always best off by not investing whatsoever, yet that *attacker groups* are better off attacking (with the right force at the right time). The reviewer’s observation that oxytocin increases social acuity and intuition, combined with our result that oxytocin increases efficient group-level attack, suggests that the oxytocin-enhanced social acuity/intuition is “used” to align individual behavior with those of others within one’s group, thus producing (unintentionally perhaps) enhanced coordination and more efficient collective action. In our revision we toned down any strong statements about the precise mechanisms underlying our observation that oxytocin leads attacker groups to be more efficient.

Also, I am not convinced that this study demonstrated that social bonding and performing synchronization rituals helps coordinate attacks (as put forth in paragraph two of the Discussion) because as I understand it, in this study group members did not interact or socially bond before coordinating their attacks. Overall, I was extremely impressed with the caliber of this thought-provoking manuscript and believe that is positively contributes to the field and scientific community.

We thank the reviewer for this helpful suggestion. We now toned down allusions to the relationship between coordinated attacks and social bonding, and revised the Discussion accordingly (paragraph two). We maintained references to work showing that attacker groups often engage in social bonding practices, and that social bonding and synchronized action can trigger the release of oxytocin. From these isolated lines of work, we derive the hypothesis that oxytocin may facilitate coordinated attacks. But, per the reviewer’s correct observation, we refrain from suggesting that social bonding increases coordinated attack through enhanced levels of oxytocin. This indeed is too speculative.

Minor Comments:Many of the figures legends are lacking definitions of the symbols used to denote significance (i.e. what significance level). Also the key is missing in Figure 5—figure supplement 1.

We thank the reviewer for this observation and now provide the information in the revised figure legend.

Discussion, third paragraph final sentence: should it be *not* needed?

Thanks and corrected accordingly.

Discussion paragraph four: "remains" should be *remained* past tense because it does not remain with the current study.

Thanks and corrected accordingly.

Reviewer #3:

This paper demonstrated that in an inter-group conflict game, exogenous oxytocin (OT) treatment increased attacker's within-group synchronization of contributions (reduced within-group variance). These behavioral changes would be because of selecting more effective strategy (attacking at a proper time when the defender was weak). This within-group coordination in attacking can increase spoils from winning. The aim of this study was based on their previous studies and was clear, and the methods were well-conducted. The number of the participants were well-enough to draw the conclusions. However, I have some concerns/questions on this form of the manuscript.

We appreciate the positive evaluation and helpful suggestions from the reviewer.

1) In Figure 1A, the authors mentioned that contribution amount was low in the subjects in the OT Condition. However, the authors already reported that OT promotes internal group defense (as a review De Dreu and Kret 2016 https://www.ncbi.nlm.nih.gov/pubmed/25908497). Therefore, it is necessary to present the data comparing Contribution amounts among four levels (Treatment × Role). If there was inconsistency between their previous study and present results regarding to the effects of OT on contribution in defense group, the authors need to discuss this.

The reviewer is correct that earlier work (reviewed in De Dreu and Kret, 2016) showed that administering oxytocin increases defensive hostility towards threatening outsiders. We find no evidence for this in the current economic contest, which mimic earlier findings that in the attacker-defender game oxytocin influences investments in attack but not in defense (De Dreu et al., 2015). To appreciate this difference in research findings it is important to note that (a) the finding that oxytocin can enhance defense hostility is obtained in individual, one-shot games where participants do not have a shared history or future, and (b) in re-iterated attacker-defender contests as studied here, defense is tough and heavily anchored on the attacker’s history of play, thus leaving little “room” for increased defense under oxytocin (compared to placebo). Thus, our null-finding for defense may be due to a ceiling effect and/or an overshadowing by attacker history. We have revised the Introduction to clarify the possible differences in setting between earlier studies finding such an oxytocin-enhanced defensiveness, and the current set of results.

– Regarding comparing Contributions among four levels (Treatment × Role):

We thank the reviewer for this enquiry; yet believe that our manuscript did provide the necessary break-downs to appreciate the significant comparisons. To be sure, we performed our ANOVA on 2 (Treatment: Oxytocin vs. Placebo) × 2 (Role: Attack vs. Defense) × 2 (Procedure: Simultaneous vs. Sequential) × 15 (Rounds), which thus fully breaks down possible comparisons. We reported significant effects of main/interaction and provided the means and statistics (Results paragraph three).

To accommodate the reviewer, we also analyzed the contributions in a 2 Treatment (Oxytocin vs. Placebo) x2 Role (Defense vs. Attack) ANOVA (thus collapsing across rounds and decision-procedure). This analysis essentially replicates the findings reported in the Main Text: Defenders invested more than attackers (F (1,78) = 287.912, p < 0.001, η^2^ = 0.787), and individuals contributed somewhat less when given oxytocin than placebo (F(1, 78) = 3.719, p = 0.057, η^2^ = 0.046; marginal significance). The Role x Treatment interaction was not significant (F<1). For attacker groups, oxytocin decreased contributions (F (1, 78) = 4.367, p = 0.040; η^2^ = 0.053) compared to placebo. No significant treatment effect was found for defender groups (F (1, 78) = 2.006, p = 0.168, η^2^ = 0.025). We believe that this 2 x 2 analysis is, however, redundant with the full model shown in the Main Text. Accordingly, we have not incorporated it in the revision at present but are happy to reconsider.

2) In Figure 1B, the data were inconsistent with De Dreu et al., 2016, in that there is no difference in the frequency of Non-contributors between Simultaneous and Sequential. The authors need to explain the inconsistency.

We thank the reviewer for pointing this out. Indeed, inconsistent with previous study (De Dreu et al., 2016) showing higher numbers of non-contributors in the simultaneous than sequential block, we did not observe a significant effect of Procedure on non-contributors in the current study. One possibility is that the previous study (De Dreu et al., 2016) used fewer rounds (5 rounds) than the current study (15 rounds). Another possibility is chance (as the patterns of the other indices, including success rate, within-group variance, and contribution, were consistent with previous findings in De Dreu et al., 2016). This possibility is not unlikely, as we did replicate the De Dreu and colleagues’ findings of decision-protocol on non-contributors in an early independent pilot study with Chinese participants.

3) In the same figure, in the Attack group of the Simultaneous and OT condition, did the frequency of Non-contributors increase in rounds in which attacks fail, and did the frequency of Non-contributors decrease when the attack succeeded? The information on the frequency of non-contributors when dividing by attacking success / failure is necessary.

We’re very grateful to the reviewer for this thoughtful suggestion. Following the reviewer’s suggestion, we conducted further analyses and included the results in the revised manuscript (Results paragraph seven).

4) If OT changes the attack efficiency, it is thought that information on how the defense parameter α has a negative regression coefficient to influence the success rate of attack is necessary.

We agree that this helps to better understand attack efficiency. We thus conducted correlation analyses between the tracking parameter α and success rate, and found that α was not correlated with success rate (r = 0.124; p = 0.276). This suggests that the oxytocin-increased attack efficiency was reflected more in terms of increasing group earnings rather than attack success. This is due to that, in the current game, winning is not necessarily associated with more earnings for attackers (r = 0.065; p = 0.565), as when the defenders contributed a lot to defend (e.g., G_Defender ≥_ 30), attackers would gain less even if they win (attack succeeds but earned leftover + spoil < 20 MU) than deciding to not contribute (attack failed but earned 20 MU).

5) In Discussion, the authors discussed one possibility that in-group defense was tacitly well-coordinated because of the stronger alignment of individual interests within defender groups. If this is true, would the results change by modulating the ratio of losing/gaining MU as the results of attach/defense? How did the authors decide the ratio of cost/benefit of the MU after the challenge?

We thank the reviewer for this question. The point that “in-group defense is tacitly well-coordinated because of the stronger alignment of individual interests within defender groups” is based on the rule of attacker and defender in the current game, as detailed in Table 1. Thus, individuals in the defender group share a stronger common fate in that when in-group defense fails, all group members gain 0 regardless of whether they contributed or not. We agree that different cost/benefit ratios and different losing/gaining ratios can alter investment behavior substantially, both for game-theoretic reasons and because of differential psychological mechanisms coming into play. It may be, though here we have to speculate, that changing the “all-or-nothing” pay-off structure for defenders to a “survive well or less well” structure may provide more room for oxytocin to influence contributions to defense. It would be interesting to examine such possibilities in future research.

Reviewer #4:

In this study, the authors examined the effects of oxytocin administration on a large sample of Chinese men (n = 480) engaged in a three-person monitory game divided to "attackers" and "defenders" where the goal was to evaluate donation to the group monetary pool in relation to the donation of own group and opponent group.It was found that oxytocin reduced donation to the group pool in the "attacker" group and increased the "coordination" between the participants' donation, defined as the inter-subject correlation between the parents' contribution. It also increased the "attackers" ability to track the opponents' contribution in sequential trials.The large sample size is an asset of the study. It is, to my knowledge, the largest oxytocin administration study conducted so far. The game is complex and so are the statistical analyses.The major issue with the study, in my opinion, is whether it measures what it claims to show. It is unclear whether the economical game the authors developed can provide any simulation to real-life battle, "warrior games" or "coordinated action" (the theoretical frame they set for the study), whether inter-subject correlations between monetary contributions of individuals and reduced variance indeed show "group synchrony" in the face of external danger, whether survival has anything to do with the neurobiological processes that underpin this game, and whether there are alternative explanations to the findings that are more plausible and parsimonious (and more consistent with findings from studies on oxytocin administration and economical games).

We appreciate the reviewer’s careful reading and apt comments regarding the framing of our work. We have taken it to heart and substantially revised the text to avoid off-hand allusions to the battle-field and to keep our revision stays closer to the data; we thank the reviewer for pointing us towards a less florid and more succinct reporting.

In terms of "coordinated action" or "behavioral synchrony" among groups, the increased group cohesion in the face of danger, and the rites and rituals group perform before going to battle, these are indeed very powerful mechanisms and are found across human cultures and non-human primates. What is unclear is how coordinated action patterns relate to contributing more or less to the general pool in a sequential economic game. There are ways to measure action coordination in groups, in terms of real action patterns, mimicry, imitation, facial expressions, coordinated movement etc, and the increase in coordination results from various factors in the group, for instance, the presence of a charismatic leader (Gordon, I. Berson, Y. Oxytocin modulates charismatic influence in groups. J Exp Psychol Gen. 2018 Jan;147(1):132-138. doi: 10.1037/xge0000375). It is unclear at all why the authors measure "behavioral synchrony" as the reduced variance among the contributions of members and it is not possible, in my opinion, to simply link this reduced variable to the literature on behavioral synchrony, action coordination, and war rituals without some kind of validation that this reduced variance in the differences of contribution over trials is indeed related to more behavioral coordination. Without such indication, it is not possible to propose that oxytocin, which is indeed a mechanism to enhance coordination of behavior (movements, vocalizations, mimicry) in humans and non-human primates, to the present findings.

We agree that coordination and synchronization occurs at various levels (neural, behavioral) and we have rewritten our manuscript to be more clear what we meant when referring to coordination – behavioral alignment of contributions to the group pool. Our definition and operationalization stays close to the work in experimental economics and games, and we hope this is now clarified. In addition, we now refrain from using the term synchrony (and also no longer reference war dances and oxytocin-induced neural synchronization studies).

An additional difficulty is the claim that literature on survival is relevant to this game in any way and that playing this game simulates mechanisms of survival. Here, I was unclear about the difference between "attackers" and "defenders" on a biological level. It appears that one group was told that they "attack" something and another that they "defend" something but attack and defend where not part of the ongoing game, even not at the level of a simple board game where there is a simulation of "defending a castle", but that the real game, for the two groups, was identical was about how to outsmart the other group in terms of donating more or less. The distance between this and proposing that the two groups differ on biological mechanisms is substantial, and must be demonstrated prior to conducting this study. It is more parsimonious (and plausible) to conclude that the mechanisms underpinning this game related more to human competition and are supported by humans' large prefrontal abilities, working memory, and far-sighted planning, and are unrelated to "movement coordination" in the face of a threat to survival. Such mechanisms are likely to be identical for the attackers and defenders (unless otherwise proven) and probably have much less to do with more primitive mechanisms described in the animal literature. For instance, the primate study the authors cite (Summani) showed greater vocal coordination and greater urinary oxytocin in primates during live conflict and is too great a leap to hypothesize from this to reduced variance of donation among players in a safe room. This study also did not differentiate between the behavioral repertoire (and oxytocin production) of attack (where motion and vocal synchrony correlated with oxytocin) and those related to "border patrol", which suggests that in the animal kingdom, both attack and defense are part of the survival and may not be biologically distinct.

As noted, we have toned down substantially our reference to these literatures, and we agree that economic contests in which one can end-up victorious over others (and win money), or being defeated by others (and lose one’s money) are hyper-stylized models of attacker-defender conflicts found in real life. Yet at the same time, our work subscribes to a strong line of work, grounded in game-theory and experimental economics, that proceeds on the basis of the assumption that such stylized models of competition and conflict can reveal understanding of the mechanistic underpinnings of strategic choices in conflict situations, including the possible neurohormonal substrates of attack and defense.

We wish to note further that participants were never told to ‘attack’ or ‘defend’. All the experimental instructions used neutral language (e.g., contribution was labeled investment; defense and attack were avoided, and groups were labeled to as group A or B). In contrast, the role of ‘attack’ and ‘defender’ is based on the contest functions in the IADC game (as now detailed in Table 1 in the revision). Individuals in the defender group share a stronger common fate in that when in-group defense fails, all group members gain 0 regardless of whether they contributed or not. Only if defense succeeds, individuals in the defender group ‘survive’ from left with 0. In contrast, attackers can always keep their remaining endowment (leftover) and spoils are additionally added into their earnings if attacks succeed. The working assumption is that these different contest success functions for attacker and defender groups elicit distinctly different social, psychological and, therefore, neurobiological operations that may be informative about “real life” intergroup competition and conflict outside the laboratory. For example, attackers are driven by the motivation to maximize gain and defenders are motivated to avoid loss, thus attack and defense would recruit distinct neurobiological systems (Nelson and Trainor, 2007; Yamagishi and Mifune, 2016). Earlier related work, and the current project, aims to uncover these possibilities. We have revised the manuscript to ensure that unwarranted generalizations are avoided.

The inter-subject correlation – the authors' index for behavioral synchrony – was measured only in the "attacker" group (rationale not clear) but this group also markedly reduced its contribution over time under oxytocin. It is possible that the increase in correlation among group members resulted from that decrease and that the zero donation correlated among the participants (even if we accept the suggestion that this is a proxy for synchrony, it is a synchrony of "no action" rather than of action).

We thank the reviewer for this question. First, we would like to clarify that the ICC was measured for both attacker and defender groups, not only for attacker group (please see Figure 5—figure supplement 1 in the revision). Second, to examine the interesting possibility that the increased ICC was driven by the number of non-contributing attackers, we calculated the correlation between ICC/within-group variance and the number of non-contributing attackers. If the increases in group coordination were driven by the number of non-contributors, we would see significant correlations between ICC/within-group variance and the number of non-contributing attackers. However, neither the ICC nor the within-group variance was correlated with the number of non-contributors (within-group variance: r =-0.073; p=0.517; ICC: r =-0.146, p=0.197). Thus the increased within-group coordination is not simply driven by the increased number of non-contributors. This null-finding is consistent with our observation that coordination does correlate with attack efficiency (i.e., greater spoils when victorious; more left-overs when attacks failed).

A more parsimonious explanation for the findings – and one that must be ruled out before offering a mechanistic explanation based on rituals and action coordination – would be that oxytocin increased the participants' memory for the sequence of events (there is evidence for oxytocin effects on memory) or that it reduced their level of stress and increased calmness and lethargy (there evidence in both humans and animals) and they reduced their contributions.

We’re grateful to the reviewer for pointing out other possible explanations of the oxytocin effect (such as the effect of oxytocin on memory or mood). If the observed oxytocin effects on the IADC game were driven by its effect on memory or mood, we would expect similar effects of oxytocin on attacker and defender groups, and similar oxytocin effects for simultaneous and sequential blocks. However, the effects of oxytocin on increased coordination, non-contributor and strategic tracking were modulated by Role and/or Procedure. Specifically, if the oxytocin-increased strategic tracking mainly reflected the effect of oxytocin on improving participants’ working memory of one’s own or in-group members’ or rival’s contributions, we would expect a generic effect of oxytocin on the tracking parameter. However, we showed a significant modulation of Procedure on the oxytocin effect on tracking (as oxytocin only increased attacker’s tracking when decisions were made simultaneously rather than sequentially). In addition, no effect of oxytocin was observed in defender groups’ tracking on rival’s history.

Regarding the potential explanation of oxytocin effects on reducing stress or increasing calmness, we conducted analysis on the mood changes (please see *SI*). Participants completed the Positive and Negative Affect Scale (PANAS) upon arrival and after the experiment to quantify their mood changes. PANAS measured both positive and negative mood that related to stress and calmness, such as nervous, distressed, alert, afraid, upset, excited, jittery, active. There was no significant mood change overall and no significant interaction effects with Role or Treatment. We also conducted the paired t-test (before- vs. after-experiment mood) for each group to examine mood change for each group. None of the group showed significant change in positive mood or negative mood (ps > 0.05,Supplementary file 1, Supplementary Tables 1C). Moreover, ANOVAs with Treatment and Role as between-subjects factors on the positive and negative mood changes, as well as overall mood changes, also did not show significant main effects of Treatment and Role, or Treatment x Role interaction (ps > 0.05).

Even within the literature on oxytocin and economic games (which should provide the conceptual frame for this study) the findings need to be reconciled with those showing that individuals donate more under oxytocin. Is this a cultural phenomenon specific to the Chinese culture?

We thank the reviewer for this question, and accordingly discussed the effects of oxytocin in the current study and in the oxytocin literature.

The variable nature of oxytocin effects on prosociality has been increasingly recognized in the oxytocin literature. Oxytocin does not simply and always promote prosociality, it has been implicated in many social behaviors, from promoting trust, generosity and cooperation (Aydogan et al., 2017; Arueti et al., 2013; Yan et al., 2018) to aggravating envy, mistrust and aggressive behavior (Ne'eman et al., 2016; Shamay-Tsoory et al., 2009; Radke et al., 2012). The seemingly contradictory oxytocin effects may be moderated by individual and contextual differences, with group context as one prominent candidate (De Dreu et al., 2010; 2011; Ma et al., 2015; De Dreu and Kret, 2016).

The current findings of oxytocin-increased group coordination, non-contributing attacker and tracking on rival’s history were consistent with the oxytocin literature that showed the effect of oxytocin on facilitating group-serving bias (De Dreu and Kret, 2016; De Dreu, et al., 2011; Ma et al., 2015), on increasing behavioral coordination and neural synchronization (Arueti et al., 2013; Mu et al., 2016), on promoting social learning and cognitive flexibility (Ma et al., 2016a; Sala, et al., 2011), and increasing salience of social feedback (Shamay-Tsoory, et al., 2016). In the revision we note and reference these literatures. While we believe we provide a nuanced review, we would be happy to further include references to studies the reviewer believes are critical to cite too.

Another important issue is the between-subject design, which is not the golden standard in oxytocin administration research. Particularly in relation to economic games, a recent study showed different brain activation patterns in a within- and between-subject design in an OT administration study using the prisoner's dilemma (Chen X, Gautam P, Haroon E, Rilling JK Within vs. between-subject effects of intranasal oxytocin on the neural response to cooperative and non-cooperative social interactions. Psychoneuroendocrinology. 2017 78:22-30. doi: 10.1016/j.psyneuen. 2017.01.006). At least, the authors should describe how randomization was conducted and show extensive comparisons among groups not only in demographics but also in relevant physiological variables (for instance, BMI).

We beg to disagree with the reviewer that between-subjects designs are deviating from the golden standard in oxytocin studies, and believe biases in randomization cannot account for our results. First, we now described the randomization procedure more fully (subsection “Procedure”), emphasizing that participants were measured not only on demographics, but also on current mood and psychological variables. We showed that individuals in the four groups (Attacker/Defender x Oxytocin/Placebo) did not differ in demographic information, mood change, and prosocial-related traits (Supplementary file 1. Supplementary Tables 1A, 1B). Second, several meta-analysis studies have summarized the behavioral (Van IJzendoorn, Bakermans-Kranenburg, 2012; Shahrestani, et al., 2013) and neural (Wang, et al., 2017) effects of oxytocin. For example, in the meta-analysis of oxytocin behavioral effects (van IJzendoorn and Bakermans-Kranenburg, 2012, Table 1), among the 31 studies examining the effects of intranasal oxytocin on face recognition, trust to in-group, and trust to out-group members, 10 studies adopted within-subjects design whereas the majority (i.e., 21 studies) employed between-subjects designs for oxytocin and placebo administration. Shahrestani and colleague summarized 7 studies examining effects of intranasal oxytocin on emotion and face recognition, all of which employed between-subjects design (Shahrestani, et al., 2013). Similarly, we summarized the neural effects of intranasal oxytocin in our recent meta-analysis of oxytocin-fMRI studies published before March 2017 (Wang et al., 2017). There were 50 fMRI studies intranasally administering oxytocin in healthy individuals, of which 29 employed between-subject design (21 studies employed within-subject design, Table S1 in Wang et al., 2017). We believe that between-subject designs is often used and producing reliable and replicable results for the oxytocin/placebo contrast, as we found here too.

In sum, the authors have collected an impressive sample and the study is complex and well-analyzed. The conceptual frame should fit the study design and alternative explanation must be carefully considered before contribution of the findings can be considered. As frames, the conclusions of the study are not supported by the data and design and may be misleading.Additional data files and statistical comments:No need for more data. The study does not measure behavioral synchrony under conflict and there is a great gap between measuring the degree of sameness in how much individuals contributed (or didn’t contribute) and the survival evolutionary frame, war rituals, and action coordination the authors invoke to support the findings. In the current frame and design, the conclusions are misleading.

We hope that our revisions clarify what we did, and what we can conclude from our results. We thank the reviewer, once again, for challenging us to provide a crisper and more factual manuscript.